# Allosteric control of the bacterial ClpC/ClpP protease and its hijacking by antibacterial peptides

Timo Jenne [1,6], Lisa Engelhardt [2,6], Ieva Baronaite [1], Dorit Levy[3], Inbal Riven[3], Maciej Malolepszy [2], Stavros Azinas [2], Taras Sych[4], Erdinc Sezgin[4], Dirk Flemming [5], Irmgard Sinning [5], Gilad Haran [3], Marta Carroni [2✉] & Axel Mogk [1✉]

## Abstract

**The hexameric AAA+ protein ClpC, combined with peptidase ClpP, forms a critical ATP-dependent protease in bacteria, essential for virulence. ClpC is usually repressed in an inactive resting state, where two ClpC spirals interact via coiled-coil M-domains. Antibacterial peptides and partner proteins trigger ClpC activation by binding to its N-terminal domain (NTD). This study reveals that the NTD stabilizes the resting state through multiple anchoring points to M-domains and ATPase domains. The same NTD sites also serve as binding sites for adaptor proteins and substrates carrying phosphorylated arginines (pArg), disrupting resting state interactions and promoting active ClpC hexamer formation. This coupling ensures that ClpC activation aligns with substrate and partner protein availability. Toxic peptides exploit this regulatory mechanism, leading to continuous ClpC activation and harmful, uncontrolled proteolysis. These findings highlight the dual role of the NTD in maintaining resting state stability and mediating activation, emphasizing its critical role in bacterial protease regulation and its potential as a drug target.**

**Keywords** Protein Degradation; ATPase Associated with Diverse Cellular Activities (AAA); Hsp100; Chaperone; Protein Quality Control
**Subject Categories** Microbiology, Virology & Host Pathogen Interaction; Post-translational Modifications & Proteolysis

## Introduction

The AAA+ protein ClpC and the peptidase ClpP form a major bacterial proteasome with crucial functions in bacterial stress resistance and virulence (Bhandari et al, 2018; Elsholz et al, 2017). ClpC forms a hexameric ring and threads substrates under ATP consumption through its central channel into the proteolytic chamber of associated ClpP for degradation. The activity of the ClpC/ClpP complex must be tightly controlled to prevent deleterious proteolysis. This is achieved by controlling ClpC ATPase activity and ClpC-ClpP interaction. In its ground state, ClpC subunits do not form hexamers but associate to form an inactive, decameric resting state composed of two pentameric half-spirals (Fig. 1A) (Carroni et al, 2017). These half-spirals associate via head-to-head interactions of coiled-coil M-domains (MD). Abrogating MD interactions by mutagenesis triggers hexamer formation and constitutive ClpC activation that is toxic to cells, underlining the crucial need for ClpC activity control (Carroni et al, 2017). This regulatory mechanism has been documented for ClpC homologs from diverse bacterial species including the major pathogens *Staphylococcus aureus* and *Mycobacterium tuberculosis* (Carroni et al, 2017; Morreale et al, 2022; Taylor et al, 2022).

ClpC can be activated by diverse signals including (i) partnering adaptor proteins that target and deliver specific substrates to ClpC (Kirstein et al, 2007; Schlothauer et al, 2003), (ii) substrate proteins harboring phosphorylated arginine residues (pArg) (Morreale et al, 2022; Trentini et al, 2016) and (iii) natural cyclic peptides with potent antibacterial activity (Malik and Brotz-Oesterhelt, 2017; Maurer et al, 2019) (Fig. 1A). The pathway leading to ClpC activation is only understood for the adaptor protein MecA. MecA binds to the ClpC N-terminal domain (NTD) and the tip of the MD (Wang et al, 2011). By binding to the MD, MecA shields MD-MD interactions sites, abrogates resting state formation and triggers ClpC activation. MecA can itself be degraded in the absence of substrate (Schlothauer et al, 2003). This represents a negative feedback loop limiting ClpC activation and ensuring high ClpC activity only in the presence of substrates.

How the binding of other adaptors triggers ClpC activation is not understood. The ClpC NTD also mediates the binding to MdfA (formerly called YjbA), a sporulation-specific adaptor recently discovered in *Bacillus subtilis* (Massoni et al, 2025) and the McsA/McsB complex, which phosphorylates arginine (pArg) residues in target proteins, generating pArg-substrates (Fuhrmann et al, 2009; Kirstein et al, 2007). pArg represents a degradation signal that is recognized by the ClpC NTD (Fuhrmann et al, 2009; Trentini et al, 2016). The NTD harbors two pArg binding sites and pArg-substrates can be degraded by ClpC/ClpP without direct assistance of adaptor proteins (Morreale et al, 2022; Trentini et al, 2016). Binding of an

[1]Center for Molecular Biology of Heidelberg University (ZMBH), DKFZ-ZMBH Alliance, Im Neuenheimer Feld 345, Heidelberg 69120, Germany. [2]Science for Life Laboratory, Department of Biochemistry and Biophysics, Stockholm University, Stockholm, Sweden. [3]Weizmann Institute of Science, Department of Chemical and Biological Physics, Rehovot 76100, Israel. [4]Science for Life Laboratory, Department of Women's and Children's Health, Karolinska Institutet, Stockholm, Sweden. [5]Heidelberg University Biochemistry Center (BZH), Im Neuenheimer Feld 328, Heidelberg 69120, Germany. [6]These authors contributed equally: Timo Jenne, Lisa Engelhardt. ✉E-mail: marta.carroni@scilifelab.se; a.mogk@zmbh.uni-heidelberg.de

artificial pArg-substrate triggers formation of ClpC hexamers that interact with other hexamers via their MDs to form a higher-order tetrahedral assembly (Morreale et al, 2022). How pArg-substrates disrupt resting state formation and enable ClpC hexamer formation is unclear. Finally, natural cyclic peptides bind to the NTD of *Mycobacterium tuberculosis* (Mtb) ClpC1 to deregulate its activity, causing cellular toxicity (Jagdev et al, 2023; Vasudevan et al, 2013; Wolf et al, 2019; Wolf et al, 2020). This property is currently exploited in the development of BacPROTAC molecules for targeted protein degradation in order to selectively degrade essential proteins in *Mycobacteria* as novel antibiotic strategy (Junk et al, 2024; Won et al, 2024). Peptides including cyclomarin A (CymA) typically enhance ClpC1 ATPase activity yet their effects on proteolysis are diverse and substrate specific (Gao et al, 2015; Gavrish et al, 2014; Greve et al, 2022; Hawkins et al, 2022). CymA enhances substrate degradation by ClpC/ClpP and triggers constitutive and uncontrolled proteolysis, rationalizing its toxicity (Maurer et al, 2019; Taylor et al, 2022). CymA binding also induces the formation of a supramolecular ClpC complex comprising four hexamers (Maurer et al, 2019; Taylor et al, 2022), similar to pArg-substrate triggered ClpC tetrahedrons (Morreale et al, 2022). However, CymA does not bind to the pArg binding pockets of the ClpC NTD but to an NTD hydrophobic groove (Vasudevan et al, 2013), which serves for binding of unfolded proteins by the homologous NTD of the AAA+ members ClpB and Hsp104 (Lee et al, 2017; Rosenzweig et al, 2015). These findings suggest the presence of distinct entry portals within the NTD to control ClpC activity. How these sites operate to link adaptor proteins, pArg-substrate or cyclic peptide binding to ClpC activation is not understood.

Here, we demonstrate that the ClpC NTD is crucial for resting state formation. We determine the structure of the ClpC resting state at higher resolution, revealing that the NTD provides diverse anchoring points that stabilize the inactive assembly. These contact points include the pArg binding pockets, but also binding sites for antibacterial peptides and adaptor proteins. Binding of the different activating components will thereby abrogate ClpC resting state formation. Moreover, we show that the resting state is dynamic, especially in the NTD regions, thus guaranteeing activation when needed. Our findings illuminate how the diverse input signals trigger a common pathway for ClpC activation.

## Results

### The N-terminal domain regulates ClpC activity

To probe for a regulatory role of the *Staphylococcus aureus* ClpC N-terminal domain (NTD), we deleted it in wild-type (WT) and F436A mutant backgrounds. The F436A mutation resides at the tip of the middle domain (MD) and abolishes resting state formation via abrogation of MD-MD interactions (Carroni et al, 2017). Therefore, ClpC-F436A is constitutively active and autonomously degrades the model substrate casein. We first determined ATPase activities of respective constructs in absence and presence of MecA, ClpP or ClpP and fluorescently labeled casein (FITC-casein) as substrate (Fig. 1B). ClpC-WT only displayed high ATPase activity in presence of its adaptor protein MecA, while addition of ClpP or ClpP/FITC-casein only slightly increased ATPase activity. Deleting the NTD (ΔN-ClpC) increased basal ATPase activity from 1.3 to 4 ATP/min and ATP hydrolysis was further enhanced in presence of

ClpP to 13.4 ATP/min. The elevated basal ATPase activity of ΔN-ClpC-F436A (7.8 ATP/min) was strongly enhanced in presence of ClpP (51.5 ATP/min) and was comparable to fully activated ClpC-WT in presence of MecA, ClpP and substrate FITC-casein (44.4 ATP/min). ClpP enhanced the ATPase activity of ClpC-F436A (from 5.2 ATP/min to 13.5 ATP/min) and the presence of substrate (FITC-casein) was necessary to reach ATP hydrolysis rates (35.5 ATP/min), which were comparable to ΔN-F436A + ClpP (Fig. 1B). Together, these findings indicate an important role of the NTD in downregulating ClpC ATPase activity as its deletion enhances ATP hydrolysis rates. The magnitude of this effect is depending on the particular ClpC background and most pronounced upon abrogation of MD-MD interactions.

Next, we tested how the differing ATPase activities of the ClpC mutants relate to FITC-casein degradation activities determined in presence of ClpP (Fig. 1C). Degradation of FITC-casein leads to an increase in fluorescence intensities and we calculated the % fluorescence increase/min as indirect, yet sensitive readout for proteolytic activity. ClpC-WT required the presence of MecA for efficient FITC-casein degradation, whereas ClpC-F436A showed high proteolytic activity on its own, confirming its constitutively activated state. Partially elevated ATPase activity of ΔN-ClpC did not allow for FITC-casein degradation, while ΔN-ClpC-F436A exhibited proteolytic activity, though it was reduced as compared to ClpC-F436A. The ClpC NTD harbors a hydrophobic groove that serves as casein binding site in case of the homologous ClpB NTD (Rosenzweig et al, 2015). We determined strongly reduced FITC-casein binding to ΔN-ClpC-F436A as compared to ClpC-F436A by anisotropy measurements (Appendix Fig. S1). This binding defect explains the reduced FITC-casein degradation activity of ΔN-ClpC-F436A despite its high ATPase activity in presence of ClpP.

We next determined proteolytic activities towards the substrate GFP-SsrA, which harbors the SsrA-degron for ClpC targeting. The 11-meric degron sequence is not recognized by the NTD but by the central pore of Hsp100 hexamers (Piszczek et al, 2005). ClpC-WT degraded GFP-SsrA in presence of MecA (Fig. 1D). ClpC-F436A did not degrade GFP-SsrA, indicating that its autonomous proteolytic activity is substrate-specific. GFP-SsrA, in contrast to FITC-casein, did not enhance ClpC-F436A ATPase activity, providing a rationale why the activated MD mutant cannot degrade this substrate (Fig. 1E). Notably, ΔN-ClpC-F436A was highly proficient in GFP-SsrA degradation and its activity was comparable to MecA-stimulated ClpC-WT (Fig. 1D). The NTD-independent processing of GFP-SsrA substantiates that the reduced FITC-casein degradation activity of ΔN-ClpC-F436A stems from reduced substrate interaction. ΔN-ClpC did not exhibit any proteolytic activity, underlining that the activating effect of the NTD deletion requires the additional abrogation of MD-MD contacts.

Together, these findings demonstrate a regulatory role of the ClpC NTD. The NTD downregulates ClpC ATPase activity and couples substrate presence to stimulation of ATP hydrolysis. Furthermore, the NTD controls substrate flux to ClpC/ClpP as evidenced by the gained ability of ClpC-F436A to degrade GFP-SsrA upon NTD deletion.

### The NTD is crucial for resting state formation

The regulatory function of the NTD suggests an impact on the ClpC assembly state. We first monitored oligomerization of ClpC-WT and

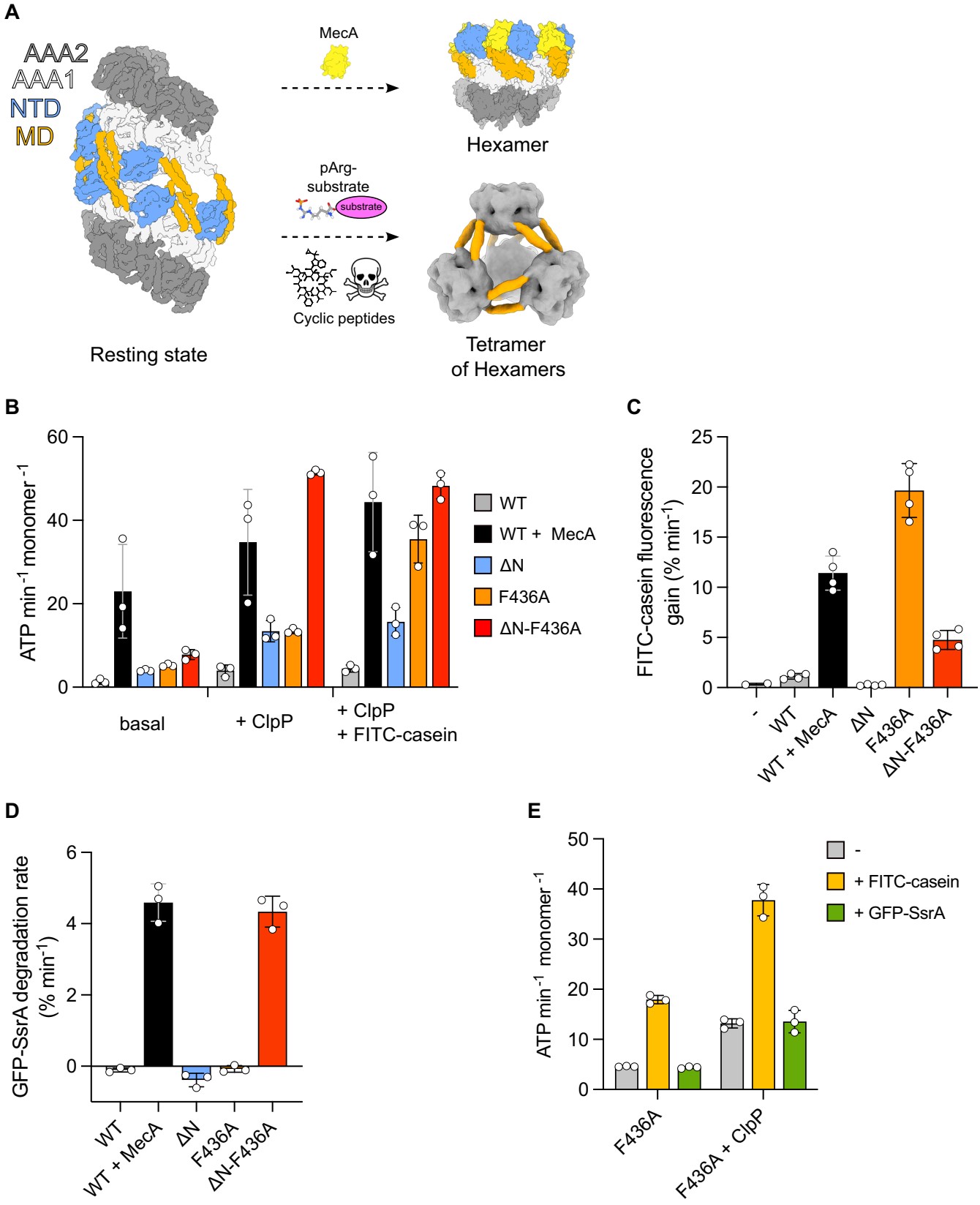

◀ **Figure 1. The ClpC N-terminal domain regulates ATPase and proteolytic activity.**

(A) Schematic representation of diverse ClpC assembly states. ClpC forms an inactive decameric resting state (pdb:6em8) and becomes active upon hexamerization via binding of adaptor proteins (e.g. MecA (pdb:3j3s)), arginine-phosphorylated substrates or toxic cyclic peptides. In the latter cases four hexamers can form a tetrahedral assembly (EMDB-11708) via MD-MD interactions. NTD N-terminal domain, MD middle domain, AAA1/AAA2 nucleotide-binding domains. (B) ATPase activities of ClpC wild-type (WT) and indicated mutants were determined in presence of MecA, ClpP and the substrate FITC-casein. (C) FITC-casein degradation activities (% fluorescence intensity increase/min) were determined in presence of ClpP, ClpC-WT and indicated ClpC mutants. (D) GFP-SsrA degradation rates (% fluorescence intensity decrease/min) were determined in presence of ClpP, ClpC-WT and indicated ClpC mutants. (E) ATPase activities of ClpC-F436A were determined in presence of substrates FITC-casein or GFP-SsrA without or with ClpP. Data information: In (B–E), column bars display mean values with standard deviations based on at least three independent experiments. Source data are available online for this figure.

mutants by glutaraldehyde crosslinking (Appendix Fig. S2A). ClpC-WT formed high molecular weight crosslink products, which hardly migrated into the SDS-gel, indicating resting state formation. Presence of MecA strongly reduced the formation of these crosslink species and a hexameric MecA-ClpC complex was formed instead, similarly to ClpC-F436A, which constitutively forms hexamers. Notably, ΔN-ClpC formed hexamers and crosslink products of intermediate sizes located between crosslinked hexamers and resting state (Appendix Fig. S2A). The formation of these assemblies involves the MD, as ΔN-ClpC-F436A forms hexamers as main crosslink product.

We next determined the sizes of ClpC assemblies by (i) dynamic light scattering (DLS) in absence and presence of ClpP and substrate FITC-casein (Fig. 2A; Appendix Fig. S2B) and (ii) by size-exclusion chromatography (SEC) (Fig. 2B). ClpC-WT assemblies had a median hydrodynamic radius of 11.2 nm in DLS measurements. The size of ClpC-WT did hardly change in presence of ClpP nor substrate and, accordingly, we did not observe complex formation between ATPase-deficient ClpC-E280A/E618A (ClpC-DWB) and ClpP in SEC runs (Fig. 2A,B; Appendix Fig. S2B). These findings reflect the formation of the inactive ClpC resting state. The presence of MecA caused formation of smaller ClpC assemblies (10.4 nm hydrodynamic radius) representing hexameric ClpC-MecA complexes (Fig. 2A). These complexes were able to interact with ClpP (Fig. 2B), forming a functional MecA-ClpC-ClpP protease with a radius of 13.0 nm. ClpC-F436A formed smaller assemblies (radius 8.1 nm) as compared to ClpC-WT, representing the active hexameric state. Accordingly, ClpC-F436A formed a proteasome-like complex with ClpP (radius 12.8 nm) (Fig. 2A,B; Appendix Fig. S2B). We conclude that the diverse ClpC assembly states and complex formation with ClpP can be faithfully monitored via DLS. ΔN-ClpC complex sizes differed from ClpC-WT and had a radius of 9.4 nm, just between the radii of resting and hexameric states. Importantly, ΔN-ClpC was able to form complexes with ClpP, in contrast to ClpC-WT. Notably, the ΔN-ClpC/ClpP complexes were of large size (median radius 18 nm) (Fig. 2A) and, accordingly, they eluted prior to canonical MecA/ClpC/ClpP or ClpC-F436A/ClpP protease complexes in SEC runs (Fig. 2B). We also observed large ΔN-ClpC-F436A/ClpP complexes, yet in presence of substrate FITC-casein, complex size was reduced to 11.4 nm, presumably representing a canonical protease complex but lacking the ClpC NTDs (Fig. 2A). Two distinct ΔN-ClpC-F436A/ClpP assemblies were also observed in SEC runs including very large complex eluting close to the void volume of the SEC column and complexes that eluted like the regular MecA/ClpC/ClpP or ClpC-F436A/ClpP complexes (Fig. 2B).

To understand the structural organizations of the diverse assemblies, we subjected them to negative staining electron microscopy (EM) (Fig. 2C). ClpC formed the resting state, while ClpC-F436A/ClpP appeared as double or single-capped protease complexes. Complexes of ΔN-ClpC/ClpP were strikingly different and appeared as long, regular rod-like structures with up to 320 nm in length (Fig. 2C; Appendix Fig. S3A). 2D class averages revealed that those structures include hexameric ΔN-ClpC rings that interact head-to-head, leaving the AAA2 rings accessible for ClpP association on both sites. ClpP also offers two docking sites for ΔN-ClpC hexamers, resulting in the formation of ΔN-ClpC/ClpP rods (Fig. 2C). The formation of ΔN-ClpC ring dimers is reminiscent of ClpL ring dimers, which also form by head-to-head MD-MD interactions (Appendix Fig. S3B), relying on a conserved aromatic residue homologous to F436A in ClpC (Appendix Fig. S3C) (Bohl et al, 2024; Kim et al, 2020). ΔN-ClpC-F436A/ClpP did not form rod-like assemblies (Fig. 2C), indicating that ΔN-ClpC ring dimer formation also relies on MD-MD contacts. EM analysis of ΔN-ClpC-F436A/ClpP showed two different types of assemblies. Smaller ΔN-ClpC-F436A/ClpP complexes represent canonical double or single-capped protease particles. Large ΔN-ClpC-F436A/ClpP complexes were composed of regular protease complexes that interacted in a complex manner. The ΔN-ClpC construct retains a flexible, 13-residue long linker sequence (147MSNKNAQASKSNN159), that might be recognized as substrate-like tail. Therefore, we assume that ΔN-ClpC-F436A/ClpP complexes might recognize other complexes unspecifically as substrate, yielding in the formation of large, non-regular assemblies. This can explain why addition of substrate FITC-casein prevents formation of such complexes by preventing ΔN-ClpC-F436A from self-recognition as substrate (Fig. 2A), which we further confirmed via SEC (Appendix Fig. S3D).

We conclude that the NTD is crucial for resting state formation. NTD deletion destabilizes the ClpC resting state and, accordingly, we determined complex formation between ΔN-ClpC and ClpP in contrast to ClpC-WT. This rationalizes enhanced ATPase activity of ΔN-ClpC in presence of ClpP. The formation of rod-like ΔN-ClpC/ClpP structures, however, obstructs proteolytic activities.

## Phosphoarginine functions as allosteric activator of ClpC

The ClpC adaptor complex McsA/McsB functions as arginine kinase and creates substrates for ClpC by phosphorylating arginine residues in target proteins (Fuhrmann et al, 2009). The ClpC NTD harbors two phosphoarginine (pArg) binding pockets (pArg1, pArg2) that mediate the recognition of respective pArg-substrates (Trentini et al, 2016). Our finding that the NTD functions as regulator of ClpC activity supports a scenario in which pArg does not only represent a recognition determinant for substrates but

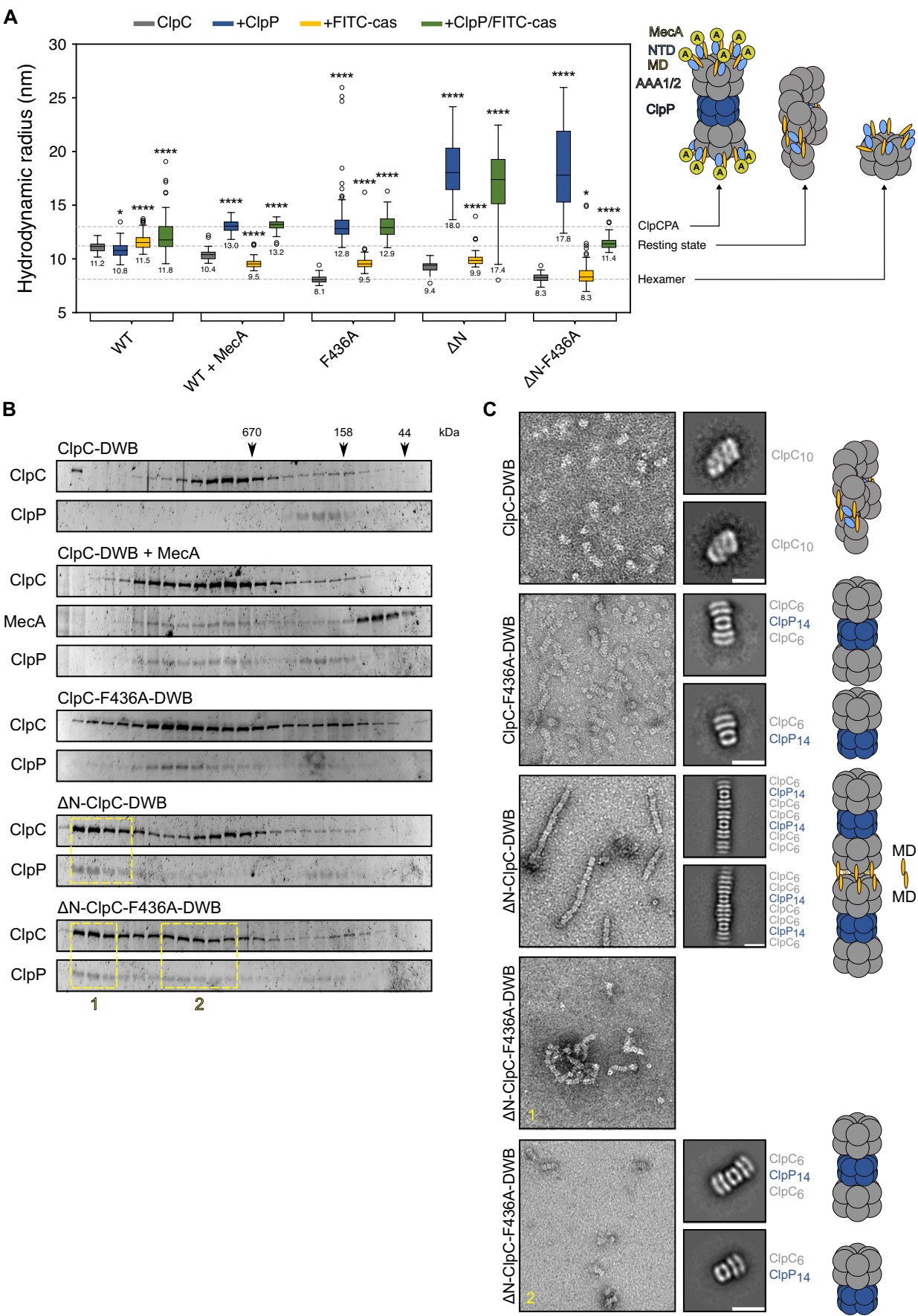

◀ **Figure 2. The ClpC NTD is crucial for resting state formation.**

(A) Hydrodynamic radii of ClpC wild-type (WT), ClpC + MecA and indicated mutants were determined by DLS measurements in absence and presence of ClpP and substrate FITC-casein as indicated (n ≥ 50). The diverse structures corresponding to the determined assembly sizes are depicted as cartoon. (ClpCPA = ClpC/ClpP/MecA). (B) Complex formation of ATPase-deficient ClpC-E280A/E618A (DWB) or indicated ClpC-DWB derivatives with ClpP was monitored by Superose 6 size-exclusion chromatography. Elution fractions were analyzed by SDS-PAGE and Sypro Ruby staining. (C) Negative stain EM of ΔN-ClpC-DWB/ClpP and ΔN-ClpC-F436A-DWB/ClpP complexes as isolated from SEC runs (see (B)). ClpC-DWB and ClpC-F436A-DWB/ClpP as reference were applied directly to the grids without prior SEC. 2D class averages reveal the formation of ΔN-ClpC-DWB/ClpP rods, composed of ΔN-ClpC-DWB hexameric ring dimers, which interact head-to-head via coiled-coil M-domains, and tetradecameric ClpP as repeating structural unit. ΔN-ClpC-F436A-DWB/ClpP form irregular assemblies including protease particles (1) and regular single and double capped protease complexes (2). The diverse structures corresponding to the observed oligomeric assemblies are depicted as cartoon. Data information: In (A), Boxes represent the interquartile range (IQR), which spans from the first quartile (Q1) to the Q3. The length of the box indicates the spread of the middle 50% of the data. Whiskers extend from the minimum and maximum values within 1.5× IQR. The median is indicated by the horizontal line. Significance of changes in particle sizes upon presence of ClpP and/or FITC-casein were determined using Brown–Forsythe and Welch one-way ANOVA tests (Unpaired t with Welch's correction). *$P$ = 0.0243 for WT + ClpP, $P$ = 0.0128 for ΔN-F436A + FITC-cas; ****$P$ < 0.0001. Source data are available online for this figure.

additionally functions as allosteric activator of ClpC. We therefore tested whether occupation of the pArg1/2 binding pockets enables ClpC-WT for processing of FITC-casein, which does not harbor phosphorylated arginine residues. Indeed, FITC-casein was degraded by ClpC/ClpP in presence of pArg at a rate that was similar to MecA-activated ClpC/ClpP (Fig. 3A). This effect was specific as activation was not observed for ΔN-ClpC/ClpP lacking the pArg1/2 binding sites or in presence of non-phosphorylated arginine (Arg) (Fig. 3A). Titration revealed an up to 10-fold stimulation of proteolytic activity in presence of 1 mM pArg and half-maximal stimulation by approx. 0.22 mM pArg (Appendix Fig. S4A). Addition of pArg enhanced ClpC ATPase activity and highest activities were determined in additional presence of ClpP and FITC-casein, reaching half of MecA-stimulated ATPase activities (Fig. 3B). The most prominent (4-fold) stimulatory effect of pArg was determined in presence of the unfolded substrate FITC-casein (Appendix Fig. S4B). This indicates that binding of substrate and pArg represents two independent signals that must function together for efficient ClpC activation. Addition of pArg to both ClpC-WT and the constitutively active hexameric variant ClpC-F436A did not allow for GFP-SsrA degradation (Appendix Fig. S4C), indicating that occupation of the pArg1/2 binding pockets is not sufficient to enable ClpC-WT or ClpC-F436A to degrade this substrate. We therefore conclude that substrate binding to the hydrophobic groove of the NTD is additionally required to overcome the repressive function of the NTD.

The ClpC NTD harbors two pArg binding sites, including E32 (pArg2) and E106 (pArg1) as crucial binding determinants (Fig. 3C). We mutated both residues (E32A, E106A) to determine the relevance of each binding site for ClpC activation by pArg. ClpC-E32A and ClpC-E106A displayed some FITC-casein degradation activity in absence of MecA, indicating partial activation that was more pronounced for ClpC-E106A (Fig. 3D). Activation was further enhanced for ClpC-E32A/E106A, whose proteolytic activity was similar to MecA-stimulated ClpC-WT (Fig. 3D). pArg enhanced proteolytic activity of ClpC-E32A but not ClpC-E106A, indicating that the pArg1 site is most relevant for ClpC activation by pArg. Accordingly, pArg strongly enhanced ATPase activity of ClpC-E32A but hardly ClpC-E106A (Appendix Fig. S4D).

pArg-bound ClpC-WT and ClpC-E32A/E106A, harboring defective pArg1/2 binding pockets, exhibit high proteolytic and increased ATPase activity. This implies that the pArg sites are important to repress ClpC activity and this function is disrupted upon pArg binding. This scenario predicts changes in the ClpC assembly state upon pArg binding or mutation of the pArg docking sites, which we monitored by DLS (Fig. 4A; Appendix Fig. S4E,F).

pArg alone hardly altered the ClpC-WT assembly size, but additional presence of FITC-casein increased it to 15.2 nm. Further addition of ClpP led to formation of very large, heterogenous complexes, which were not observed in presence of pArg/ClpP (Fig. 4A; Appendix Fig. S4E). A similar trend was observed for ClpC-E32A/E106A whose complex size was similar to ClpC-WT and did not change in presence of ClpP. However, presence of FITC-casein also increased ClpC-E32A/E106A complex size to 15.1 nm, which is highly similar to the size of ClpC-WT bound to pArg and FITC-casein (Fig. 4A; Appendix Fig. S4F). Further addition of ClpP again caused formation of very large assemblies, as observed for ClpC-WT/pArg/FITC-casein/ClpP complexes.

A ClpC assembly size of approx. 30 nm diameter is consistent with a complex formed by four ClpC hexamers, which has been described for ClpC bound to either cyclomarin A or to an engineered BacPROTAC molecule, mimicking a pArg-substrate (Maurer et al, 2019; Morreale et al, 2022). The four ClpC hexamers interact via their coiled-coil MDs forming a tetrahedral structure (Fig. 4B). The ClpP-interacting AAA2 rings are fully accessible in each ClpC hexamer of the assembly, allowing for ClpP docking. The interaction of ClpP and ClpC tetrahedrons will cause formation of a branched, large meshwork of ClpC/ClpP complexes (Fig. 4B). This is consistent with the very large and heterogenous complexes of ClpC-WT/pArg/FITC-casein (median radius 57.4 nm, range 23.7–940 nm) and ClpC-E32A/E106A (median radius 45.9 nm, range 20.14–170 nm) determined in presence of ClpP (Fig. 4A, Appendix Fig. S4E/F). The assembly size of the ClpC MD mutant F436A, which cannot undergo MD-MD interactions, remained unaltered in presence of pArg without and with ClpP (Appendix Fig. S5A), supporting that pArg and FITC-casein binding induces a ClpC tetrahedral state.

Negative staining EM analysis of ClpC-WT bound to pArg revealed a resting-state like assembly consistent with DLS data (Fig. 4C; Appendix Fig. S5C). The assemblies, however, appeared less regular and less homogenous as compared to ClpC-WT (Fig. 4C; Appendix Fig. S5B), implying alterations of resting state organization upon pArg binding. ClpC-WT bound to pArg and FITC-casein led to formation of tetrahedrons (~75% of particles), but also complexes consisting of two ClpC ring dimers (~25% of particles) could be observed (Fig. 4C; Appendix Fig. S5D). These ring dimers differed structurally from the resting state as confirmed by low-resolution 3D density maps that revealed a planar assembly

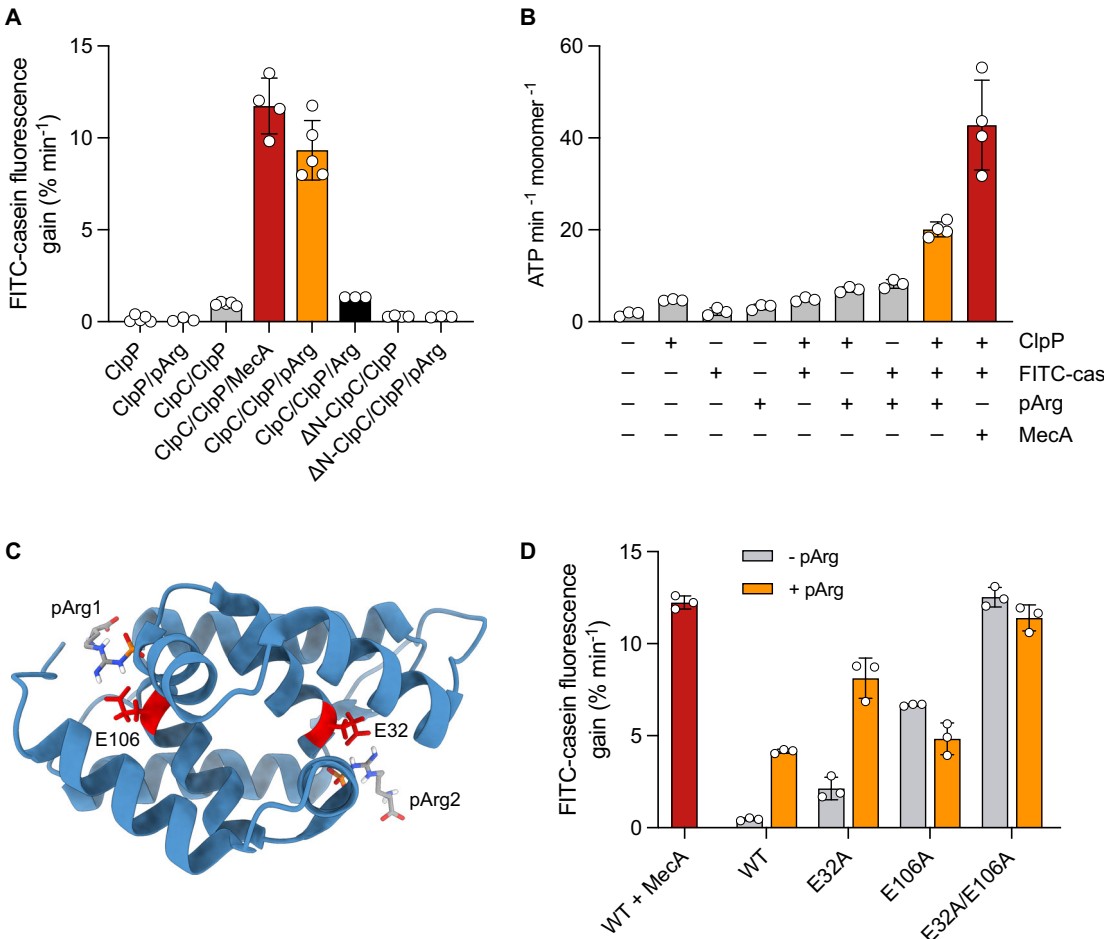

**Figure 3. Occupying or mutating pArg binding sites triggers ClpC activation.**

(A) FITC-casein degradation activities (% fluorescence intensity increase/min) were determined in presence of ClpP, ClpC wild-type and NTD-deleted ClpC (ΔN) in absence and presence of 1 mM phosphorylated arginine (pArg) or 1 mM arginine (Arg). (B) ATPase activities of ClpC wild-type in presence of ClpP, FITC-casein, MecA and 250 μM pArg were determined as indicated. (C) Structure of the isolated *B. subtilis* NTD bound to pArg (pdb:5hbn). The pArg binding sites (pArg1, pArg2) and conserved glutamate residues (E32, E106) are indicated. (D) FITC-casein degradation activities (% fluorescence intensity increase/min) of ClpC-WT and indicated mutants were determined in absence and presence of 250 μM pArg. Data information: In (A, B, D), column bars display mean values with standard deviations based on at least three independent experiments. Source data are available online for this figure.

of the two rings in contrast to the spiral arrangement of the resting state (Appendix Fig. S5E). ClpC-E32A/E106A formed complexes composed of 4–5 interacting hexamers in presence of the substrate FITC-casein, while alone it formed assemblies resembling a ClpC resting state though displaying some heterogeneity similar to pArg-bound ClpC-WT (Fig. 4D; Appendix Fig. S5F,G). Together, these findings indicate a common structural pathway for substrate-induced activation of pArg-bound ClpC-WT and ClpC-E32A/E106A leading to formation of ClpC hexamers that interact via their MDs.

We infer that ClpC-E32A/E106A resembles pArg-bound ClpC-WT, requiring substrate presence for changes in assembly and activity state. The similar biochemical characteristics of pArg-bound ClpC-WT and ClpC-E32A/E106A point to a model in which (i) pArg binding pockets provide crucial contact sites to stabilize the ClpC resting state and (ii) pArg binding to such sites abrogates these interactions causing destabilization of the resting state.

## Cryo-EM structures of ClpC show formation of multiple resting states and flexibility of the NTDs

To precisely understand how the NTD contributes to resting state formation, we re-determined the structure of ClpC-WT by cryo-electron microscopy single particle analysis (cryo-EM SPA). This was necessary as the cryo-EM model that we had previously generated (Carroni et al, 2017) had limited resolution (~8 Å) and did not allow for unambiguous placing of the NTDs. Prior to structure determination, we performed mass photometry analysis to obtain insights into the stoichiometry of the ClpC resting state as we suspected that compositional heterogeneity could have been the reason for the limited resolution. ClpC-WT formed three distinct species in presence of ATP that corresponded to complexes consisting of 10, 12 and 14 subunits (Fig. 5A). This finding was important in informing the cryo-EM data analysis and guiding us to classify and refine three distinct maps corresponding to the 10-, 12- and 14-mer at a resolution of around 3.8 Å (Fig. 5B; Appendix

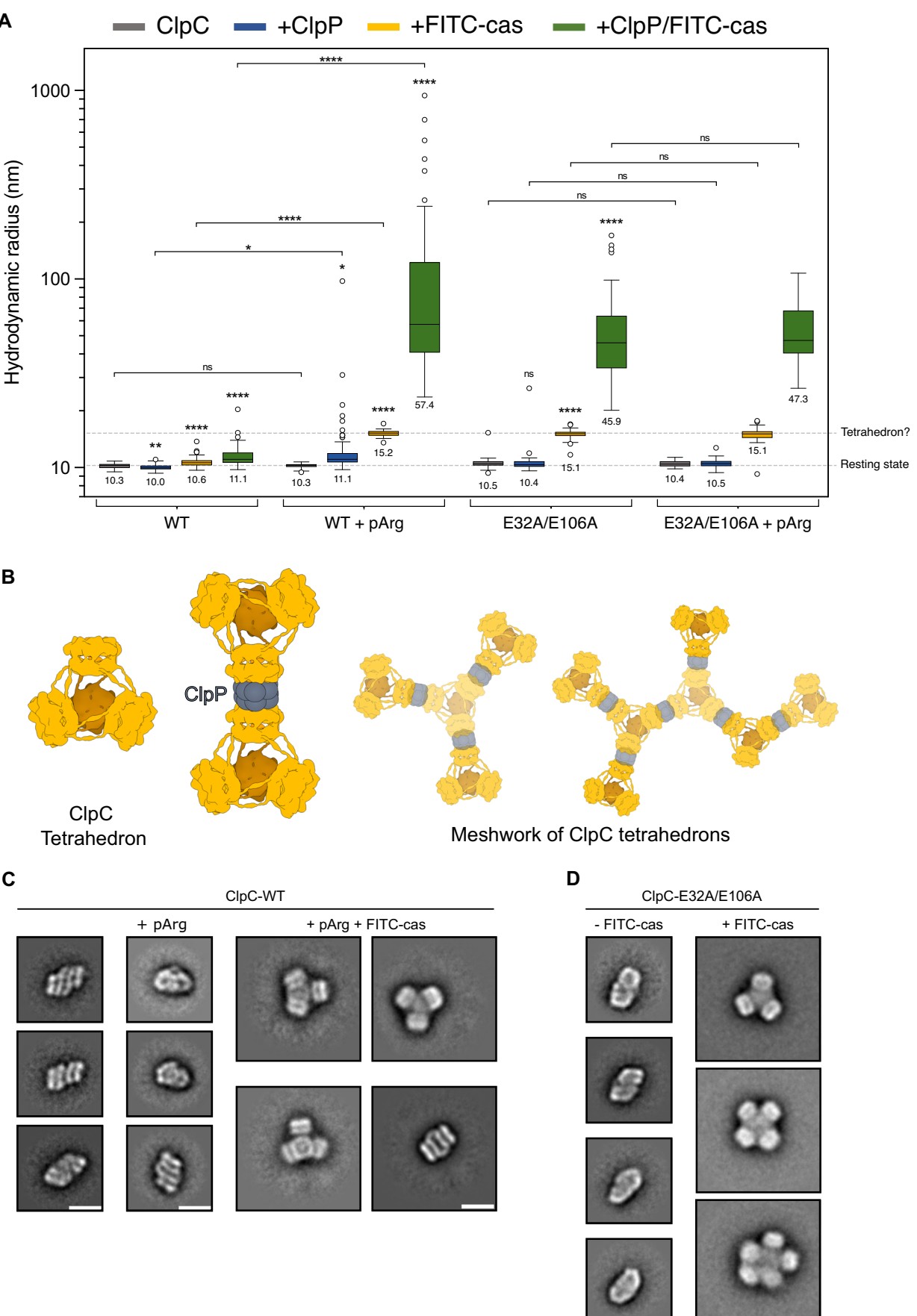

**Figure 4.   Substrate-dependent shift from resting state to hexamer-based ClpC assemblies.**

(A) Hydrodynamic radii of ClpC-WT and pArg-binding site mutant ClpC-E32A/E106A were determined by DLS in presence of ClpP and FITC-casein as indicated. (B) Tetrahedron assemblies consisting of four ClpC hexamers do interact via coiled-coil MDs. ClpP can associate with and link multiple tetrahedrons leading to formation of large meshworks of ClpC/ClpP complexes. Tetrahedron cartoon is based on EMDB-11708. (C) 2D class averages of ClpC-WT alone, bound to pArg or both pArg and FITC-casein based on negative stain EM reveal formation of tetrahedron assemblies. (D) 2D class averages of ClpC-E32A/E106A in absence or presence of FITC-casein based on negative stain EM. Substrate binding shifts the resting state into larger assemblies involving four to five interacting ClpC rings. (C, D) Scale bar = 20 nm. Data information: In (A), data are represented as described in Fig. 2. Significances of changes in particle sizes upon presence of ClpP and/or FITC-casein ± pArg were determined using Brown–Forsythe and Welch one-way ANOVA tests (Unpaired t with Welch's correction). n.s. not significant, $P = 0.701$ for WT vs. WT + pArg, $P = 0.574$ for E32A/E106A vs. E32A/E106A + ClpP, $P = 0.4065$ for E32A/E106A vs. E32A/E106A + pArg, $P = 0.4235$ for E32A/E106A + ClpP vs. E32A/E106A + ClpP + pArg, $P = 0.6656$ for E32A/E106A + FITC-cas vs. E32A/E106A + FITC-cas + pArg, $P = 0.9864$ for E32A/E106A + ClpP + FITC-cas vs. E32A/E106A + ClpP + FITC-cas + pArg; $*P = 0.0254$ for WT + pArg vs. WT + ClpP + pArg, $P = 0.018$ for WT + ClpP vs. WT + ClpP + pArg; $**P = 0.0019$ for WT + ClpP; $****P < 0.0001$ for all conditions. Source data are available online for this figure.

Fig. S6; Movie EV1). The distinct structures all represented ClpC resting states and were similarly abundant (Appendix Fig. S6C), consistent with mass photometry data. No other oligomeric intermediate was observed in mass photometry or cryo-EM SPA, suggesting that octameric or smaller species are unstable. The differences in subunit numbers of the ClpC resting states suggest that assembly or disassembly of the complexes happens by addition or removal of dimeric building blocks (Movie EV1).

The three cryo-EM structures of the ClpC resting state have the same overall double staircase head-to-head arrangement that we reported previously (Fig. 5B; Movie EV1, pdb:6em9) (Carroni et al, 2017). The resting state is kept together by several surfaces of interactions, mainly interchain (Appendix Fig. S7A). The largest interface area between adjacent subunits involves laterally the AAA1 and AAA2 domains ($\sim1600\,\text{Å}^2$) to form the helical arrangement (Appendix Fig. S7A). A $\sim940\,\text{Å}^2$ area involves contacts of NTDs with MDs ($\sim220\,\text{Å}^2$ intrasubunit and $\sim380\,\text{Å}^2$ intersubunit) and AAA1-domains ($\sim340\,\text{Å}^2$ intrasubunit) and a $\sim360\,\text{Å}^2$ area involves head-to-head MD contacts (Appendix Fig. S7A). Contacts between the MDs involve the conserved and well-characterized F436 residue, which is essential for resting state formation (Carroni et al, 2017) (Appendix Fig. S7A).

The three oligomeric states aligned to each other almost perfectly (CC of 0.89) on the "front side" of the double staircase while they differ in the "back side", which is open as a cradle in the 10-mer and closed with a seam in the 14-mer (Fig. 5B). Upon signal subtraction and local refinement of dimers from the 14-mer dataset, we obtained the highest-resolution map (3.5 Å) of a ClpC resting-state dimer (see "Methods", Appendix Fig. S6E) then merged into a 14-mer combined map. We were not able to improve the resolution any further, possibly due to inherent flexibility within each chain, especially for those in the "back side" of the structures. Accordingly, the local resolutions of back side subunits were lower as compared to those of the front side, indicating increased flexibility of this region (Fig. 5C). Furthermore, the overall resolution range increased from the 10-mer to the 12- and 14-mer resting state, suggesting increased dynamics of the 10-mer assembly (Fig. 5C).

The increased resolution of the 14-mer resting state and the generation of an improved AlphaFold2 model of the *S. aureus* ClpC monomer allowed us to partially refine the full resting state structure and to precisely position the NTD into the cryo-EM density (Appendix Table S1). Additionally, this partially enabled us to determine the nucleotide binding states of some AAA2 domains. Out of the 14 ATP-binding sites at AAA2, 8 are occupied with nucleotides (7 ATPs and 1 ADP, Appendix Fig. S7B). These are the

densities that could be clearly identified, although some residual densities are also present in the nucleotide binding pockets of other AAA1 and AAA2 domains, which however cannot be precisely assigned. While loaded, the ATPase activity of AAA2 will remain low as the highly-pitched assembly of the ATP pockets prevents allosteric arginine-fingers (R704) from neighboring subunits to access bound nucleotides (average distance 9 Å) and trigger ATP hydrolysis (Appendix Fig. S7B). Still, loaded ATPs can make the subunits promptly activatable, once their ATPase pockets assume a planar hexameric conformation.

Next, we focused our structural analysis on the NTD, as our biochemical data point to its crucial role in resting state formation. The NTDs are resolved differently in different subunits, with the ones of the "front side" being well defined and the ones of the "back side" less. Two of the subunits in the "back side" are only visible at low threshold (Appendix Fig. S8A). In "front side" subunits NTDs are sandwiched between MDs and also contact AAA1 domains. While this position is also seen in the previous structure (pdb:6em9), the NTDs are now placed differently and are rotated by 180° (Appendix Fig. S8B). Furthermore, in the former model, the NTD extended into the adjacent subunit, while in the new structure we have built the NTD on top of its own chain (Appendix Fig. S8B), which implies larger mobility of the flexible long linker connecting the NTD to the AAA1.

The sandwiched position of the front side NTDs will render them largely immobile. However, the back side subunit NTDs are more flexible and can occupy different positions (Fig. 5D; Appendix Fig. S8A) and are observed either inside or outside of the resting state chamber (Fig. 5D; Appendix Fig. S6F). Seam subunits NTDs of the 14-mer state make contact with different domains in the inner surface of the assembly (Fig. 5D; Appendix Fig. S6G; Movie EV2). Additionally, motion of these NTDs, going from the outside to the inside of the 14-mer resting assembly, is observable upon masked 3D classification (without alignment) of the region (Appendix Fig. S6F) as well as 3D variability (Movie EV3) and cryoDRGN analysis (Appendix Fig. S6F,G; Movies EV2 and3). Similarly, NTDs of "back side" subunits of the 10- and 12-mer resting states are not sandwiched between MDs but partially visible on the inner site of the complexes (Appendix Fig. S8A; Movies EV2 and 3). We infer that NTD subunits show strong differences in their mobilities depending on their localization in the resting state, with the ones on the back seam side being highly mobile. The diverse positions might allow NTDs to play distinct roles in stabilizing the resting state but also enables subunit mobilization and ClpC activation.

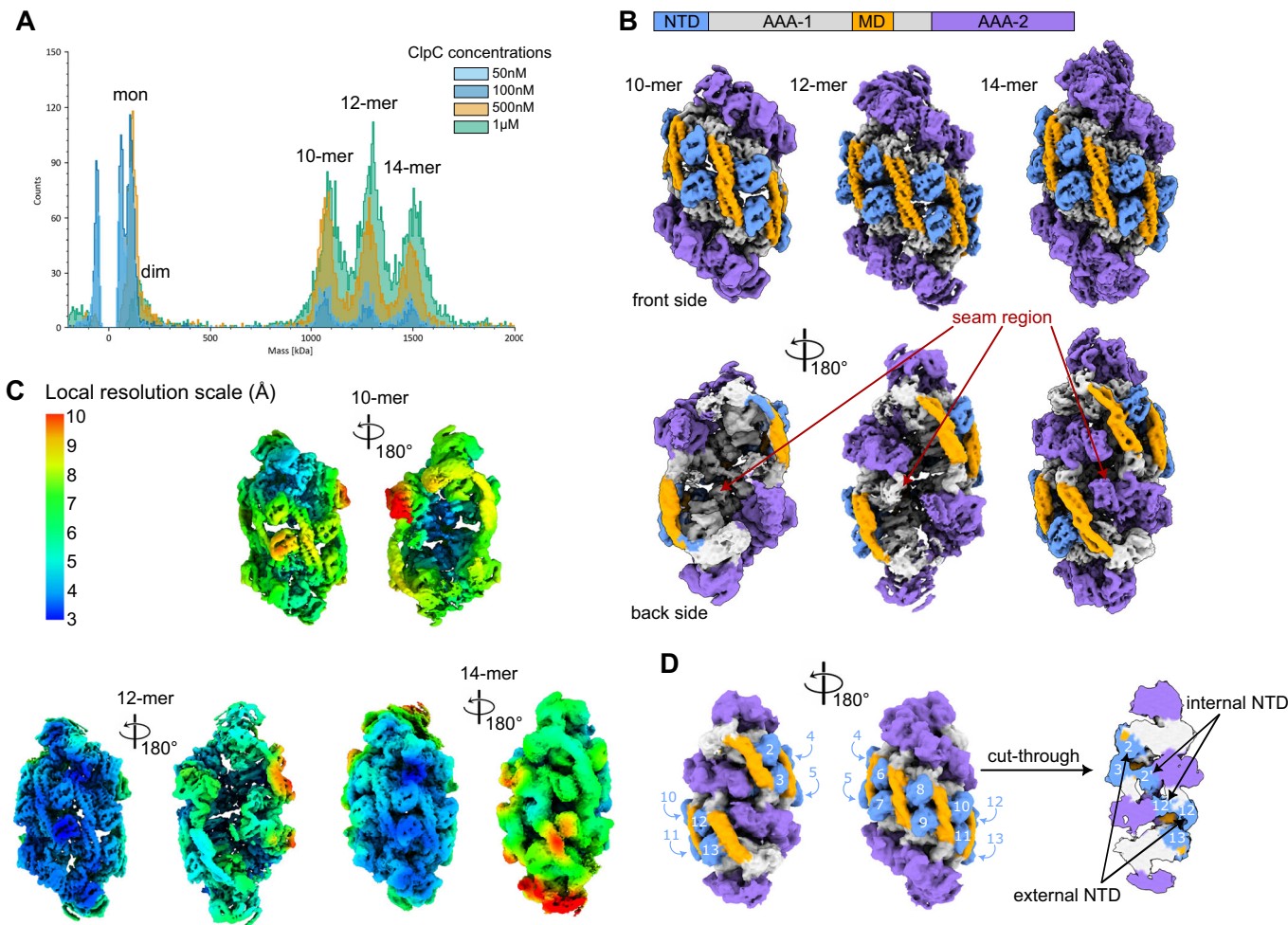

**Figure 5. ClpC resting state exists in different oligomeric states, characterized by a region of high flexibility.**

(A) Mass photometer readings of ClpC-WT at various concentrations (50 nM, 100 nM, 500 nM and 1 μM), incubated with 2 mM ATP. Different weight peaks can be observed, corresponding to monomeric and dimeric ClpC as well as oligomers of 10, 12 and 14 subunits. It has to be noted that the measurements for the monomeric and dimeric peaks are inaccurate due to high concentration of the sample. (B) Cryo-EM reconstructions of the different ClpC oligomeric resting states (10-mer, 12-mer and 14-mer). The "front" sides of the resting states are similar to each other, whilst the "back" sides vary depending on the oligomeric state. (C) Local resolution of the different oligomeric resting state structures. Resolution ranges from 3 to 10 Å, with highest resolution being in the "front" side and lowest resolution at the "back" side of the structures. The resolution is the lowest for the 10-mer volume, indicating a less stable conformation compared to the 12- and 14-mer. (D) Position of the NTDs in the 14-mer structure. 12–13 NTDs are visible in the resting state (depending on lower/higher threshold). The NTDs of the seam, back side subunits are either located externally or internally, pointing at their high mobility (see also Movies EV2 and 3).

## NTD anchoring points are crucial for resting state formation

To mechanistically understand the role of the NTD in resting state formation we analyzed NTD interactions of front side subunits. Here, NTDs are involved in various specific contacts with other domains of the same and the neighboring subunit. In particular, NTD residue K85 makes intra-subunit salt bridge interactions with residues D356 and E359 of the AAA1 (Fig. 6A), highlighting a possible regulatory contact that connects the ATPase domain with the NTD. The NTD residue R122 hydrogen bonds with N462, which resides at the base of the MD of the same subunit (Fig. 6A). Each NTD also makes extensive contacts with the MD of the adjacent subunit (MD*), mainly mediated by a patch of hydrophobic and apolar amino acids and one clear salt bridge between conserved residues R9 and E435 (Fig. 6A). Notably, R9 and

R122 are located at or near the pArg1 and pArg2 binding pockets, which are thus part of the NTD interaction network in the resting state. Indeed, T7 is part of the pArg1 site and contacts E426 of the adjacent MD* for which, upon mutation, a partial activation has been noticed before (Carroni et al, 2017) (Appendix Fig. S9A).

We probed for the functional relevance of the diverse NTD interactions within the resting state by creating ClpC mutants that are expected to weaken anchoring points between NTD-AAA1 (K85-D356), NTD-MD (R122-N462) and NTD-MD* (R9-E435 and T7). The mutants T7D, R9A, K85A, D356A and E435A exhibited MecA-independent proteolytic activity towards FITC-casein, while R122A and N462A remained dependent on MecA for substrate degradation (Fig. 6B; Appendix Fig. S9B). Proteolytic activities of ClpC-T7D, ClpC-K85A and ClpC-D356A were comparable to or even exceeding MecA-activated ClpC-WT or

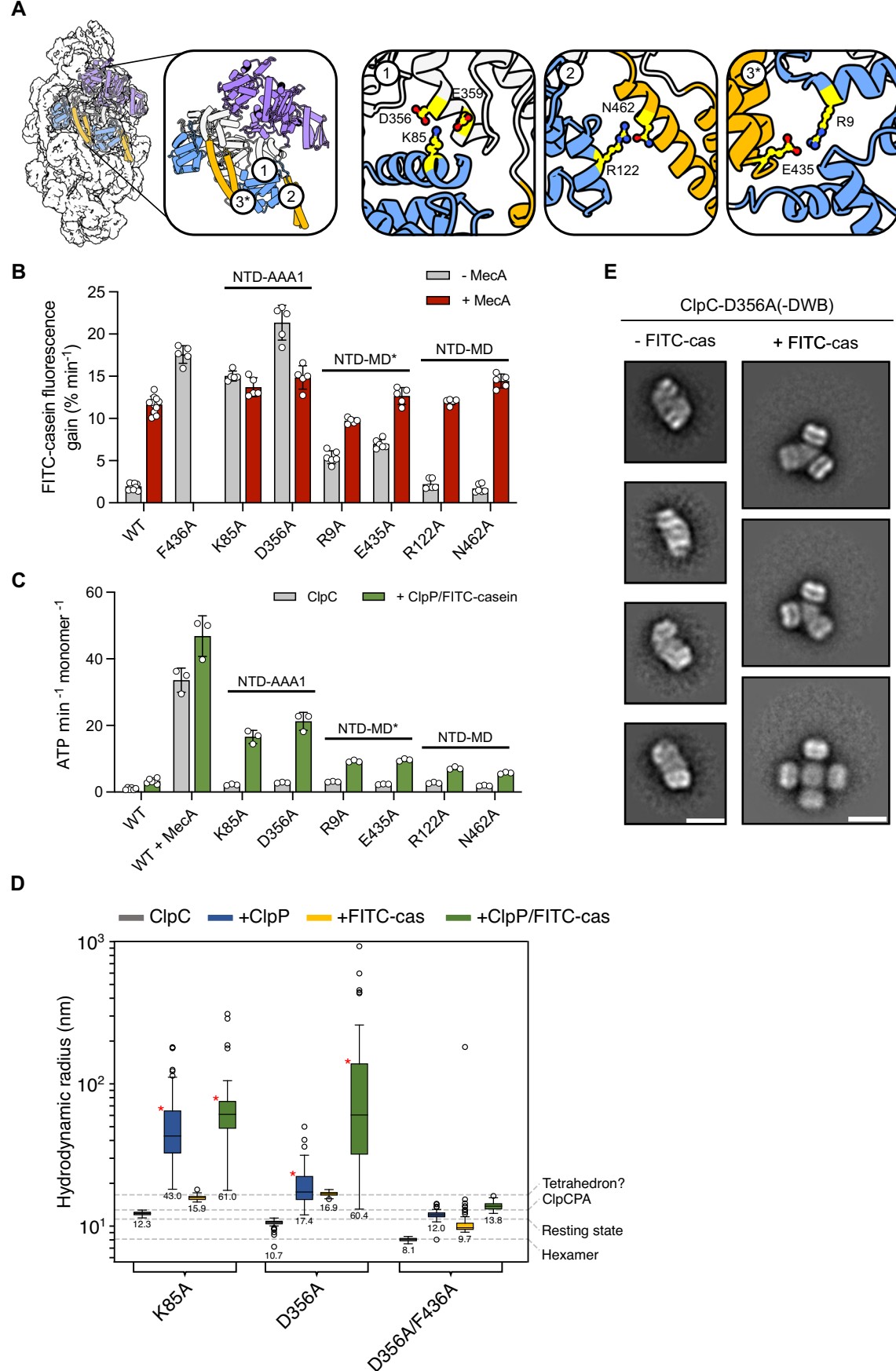

**Figure 6.  NTD anchoring points are crucial for ClpC activity control.**

(A) Two adjacent subunits within the resting state are highlighted. NTD anchoring points highlighting contributing interacting residues are displayed as followed: NTD-AAA1 (K85-D356/E359), NTD-MD (R122-N462) and NTD-MD* (neighboring MD): R9-E435. (B) FITC-casein degradation activities (% fluorescence intensity increase/min) were determined in presence of ClpP and ClpC (WT and indicated mutants) in absence and presence of MecA. The NTD anchoring points affected by the distinct mutations are indicated. (C) ATPase activities of ClpC (WT and indicated mutants) were determined in absence and presence of ClpP and FITC-casein. (D) Hydrodynamic radii of indicated ClpC mutants were determined by DLS in absence and presence of ClpP and FITC-casein as indicated. (E) 2D class averages of ClpC-D356 in absence or presence of FITC-casein based on negative stain EM. ClpC-D356A-DWB + FITC-casein complexes were first isolated by Superose 6 SEC runs. Scale bar = 20 nm. Data information: In (B, C), column bars display mean values with standard deviations based on at least three independent experiments. In (D), Boxes represent the interquartile range (IQR), which spans from the first quartile (Q1) to the Q3. The length of the box indicates the spread of the middle 50% of the data. Whiskers extend from the minimum and maximum values within 1.5× IQR. The median is indicated by the horizontal line. Red asterisks indicate heterogenous samples with multiple peaks, for which the value with the smallest hydrodynamic radius was included for data representation. Source data are available online for this figure.

ClpC-F436A, demonstrating strong activation. Addition of MecA to D356A led to a partial inhibition of FITC-casein degradation, likely through competition of MecA with FITC-casein for ClpC-mediated degradation. Activated ClpC mutants also displayed increased ATPase activities in presence of ClpP and FITC-casein, underlining their derepression (Fig. 6C; Appendix Fig. S9C). These findings indicate a dominant role of the NTD-AAA1 (K85-D356) anchoring point for resting state formation, and while the NTD-MD* (R9-E435 and T7-E426) interaction also contributes to ClpC repression, the mutant pair affecting the NTD-MD contact point (R122-N462) displayed largely WT-like activities.

We next determined the assembly states of ClpC mutants with highest autonomous activities (T7D, K85A, D356A) by DLS. Complex size of ClpC-T7D and ClpC-D356A was similar to ClpC-WT, while the radius of ClpC-K85A was slightly increased (12.3 nm vs. 10.3 nm) (Fig. 6D; Appendix Fig. S9D,E). All mutants were able to associate with ClpP, as revealed by an increase in complex radii upon peptidase addition (Fig. 6D; Appendix Fig. S9D,E). Addition of FITC-casein triggered formation of defined complexes with radii of 15.9–16.9 nm, similar to those observed before for ClpC-E32A/E106A and pArg-bound ClpC-WT (Fig. 4A). Such complexes were not formed by ClpC-D356A/F436A, again indicating that the formation of larger assemblies relies on MD-MD contacts (Fig. 6D). We confirmed the impact of bound substrate FITC-casein on ClpC-D356A assembly size by SEC analysis and subsequent fluorescence measurements of the eluted fractions (Appendix Fig. S9F). EM analysis documents that ClpC-D356A forms a resting state-like structure in absence of substrate (Fig. 6E; Appendix Fig. S9G). Addition of FITC-casein triggers formation of hexamer-based ClpC-D356A assemblies, largely representing tetramers of ClpC hexamers, but also assemblies consisting of at least five ClpC hexamers (Fig. 6E; Appendix Fig. S9G). These assembly states again explain the formation of very large, heterogenous assemblies of FITC-casein bound ClpC-D356A in additional presence of ClpP (Fig. 6D).

Together, our structural and biochemical findings determine the crucial contributions of the NTD to ClpC resting state formation. Abrogating NTD/AAA1 and NTD/MD* anchoring points destabilizes the ClpC resting state and enables for autonomous proteolytic activities. All residues contributing to these interactions are conserved in ClpC homologs (Appendix Fig. S10), indicating that the regulatory function of the NTD is conserved.

## Cyclic peptides activate Mtb ClpC1 by blocking a crucial NTD anchoring residue

Cyclic peptides including cyclomarin A (CymA) trigger cellular toxicity by binding to a hydrophobic groove of the *M. tuberculosis*

(Mtb) ClpC1 NTD (Malik and Brotz-Oesterhelt, 2017). Mtb ClpC1 is also controlled by resting state formation (Taylor et al, 2022) and, accordingly, NTD residues which we show having regulatory functions are conserved in Mtb ClpC1 (Appendix Fig. S10). We could show before that CymA overrides ClpC1 activity control by triggering formation of an active ClpC1 tetrahedron complex (Maurer et al, 2019). Such complex is highly similar to the assemblies formed by activated *S. aureus* NTD mutants (e.g. E32A/E106A) or pArg-bound ClpC-WT in additional presence of substrate (Fig. 4A,C,D). While this suggests a common pathway of ClpC activation by pArg-substrates and cyclic peptides, the peptides do not interact with the pArg1/2 binding pockets, raising the question of how they abrogate resting state formation.

While cyclic peptides bind to the hydrophobic groove of the Mtb ClpC1 NTD, they all also contact residue K85, which is crucial for resting state formation of *S. aureus* (Sa) ClpC (Figs. EV1A and 6) (Jagdev et al, 2023; Vasudevan et al, 2013; Wolf et al, 2019; Wolf et al, 2020). Superimposition of ClpC1-NTD co-crystal structures bound to cyclic peptides into the ClpC resting state revealed a steric clash with the Sa ClpC K85-D356 interaction pair (corresponding to K85/D364 in *M. tuberculosis* ClpC1) (Figs. 7A and EV1B). We probed for the roles of K85 and D364 in Mtb ClpC1 regulation by determining proteolytic and ATPase activities of ClpC1-K85A and ClpC1-D364A, which we predict to mimic a cyclic peptide-bound state. Both mutants showed strongly enhanced proteolytic activity, which was even higher than CymA-activated ClpC1-WT and similar to the MD mutant ClpC1-F444S, which constitutively forms hexamers (Taylor et al, 2022) (Fig. 7B). Similar findings were obtained when determining ATPase activities in presence of ClpP and FITC-casein: ClpC1-K85A and ClpC1-D364A exhibited increased ATPase activities, which were similar to CymA-activated ClpC1-WT (Fig. 7C). Notably, FITC-casein degradation by ClpC-K85A was partially inhibited by CymA. We speculate that CymA binding to ClpC1-K85A might be slightly altered, potentially leading to a stronger competition between CymA and FITC-casein for NTD binding in ClpC1-K85A.

We infer that the allosteric control of ClpC activity by the NTD is evolutionarily conserved. Binding of cyclic peptides to Mtb ClpC1 will abrogate NTD-AAA1 key interactions, mechanistically explaining how they deregulate ClpC1 activity and trigger cell death.

## Activated ClpC NTD mutants are toxic in vivo

Finally, we tested for the physiological relevance of NTD function in ClpC activity control by co-expressing plasmid-encoded

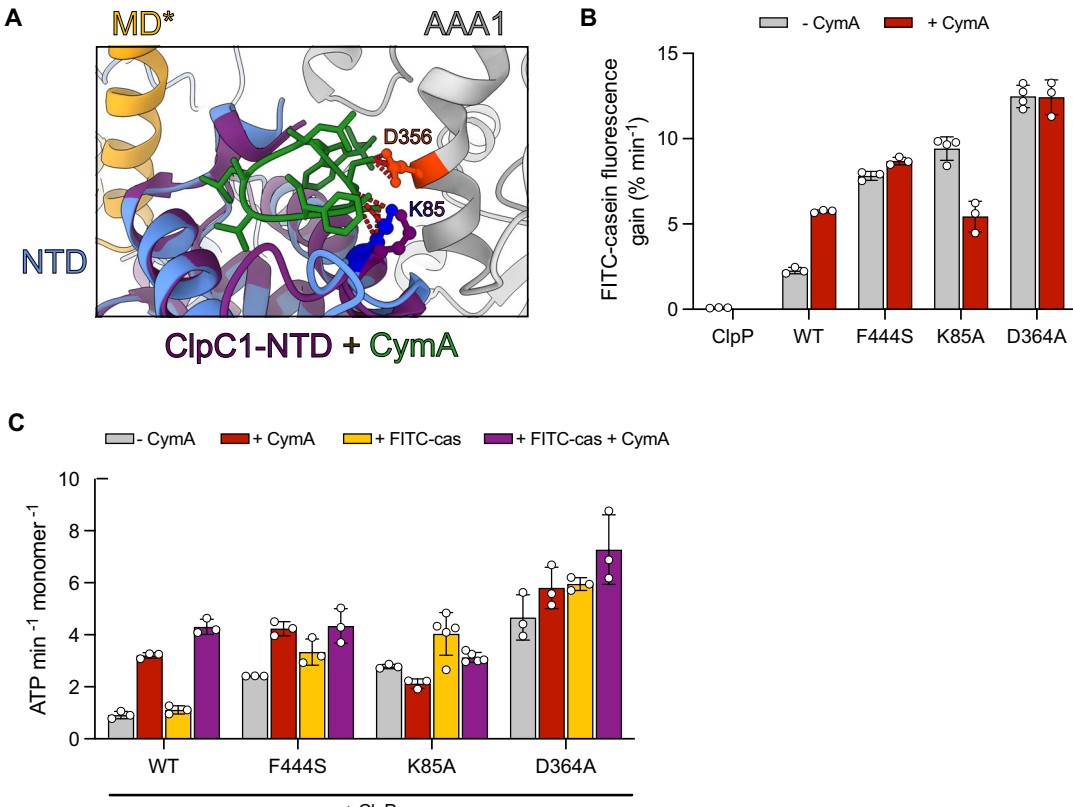

Figure 7. Cyclic peptides disrupt the NTD-AAA1 anchoring point in Mtb ClpC1.

(A) The structure of Mtb ClpC1 N-terminal domain (NTD) bound to cyclomarin A (CymA) (pdb: 3wdc) was superimposed with the *S. aureus* ClpC resting state. CymA contacts K85 and clashes with D356 (D364 in Mtb ClpC1), indicated by dark red dashed lines. (B) FITC-casein degradation activities (% fluorescence intensity increase/min) of Mtb ClpC1 wild-type (WT) and indicated mutants were determined in absence and presence of CymA as indicated. (C) ATPase activities of Mtb ClpC1-WT and indicated mutants were determined in absence and presence of CymA and FITC-casein as indicated. Data information: In (B, C), column bars display mean values with standard deviations based on at least three independent experiments. Source data are available online for this figure.

ClpC-WT or activated mutants with *S. aureus* ClpP from IPTG-controlled promoters in *E. coli* cells. We showed before that expression of ClpC-F436A but not ClpC-WT creates toxicity (Carroni et al, 2017), underlining the need for ClpC activity control in cells. ClpC mutants that displayed high autonomous proteolytic activity in vitro caused toxicity in vivo (Fig. 8; Appendix Fig. S11A). Toxic effects increased with growth temperatures and were comparable to ClpC-F436A. Levels of toxic ClpC mutants were typically lower as compared to ClpC-WT (Appendix Fig. S11B,C), excluding that toxicity is caused by increased production. Notably, activated ClpC mutants also formed truncated forms in vivo, likely lacking the NTD (Appendix Fig. S11C). This can be explained by partial autoprocessing as observed before for ClpC-F436A (Carroni et al, 2017), further confirming their activated state. These findings demonstrate the physiological relevance of ClpC activity control via the NTD and the devastating consequences upon its loss. Notably, expression of ΔN-ClpC-F436A was more toxic than ClpC-F436A (Fig. 8). This finding correlates with the gain of proteolytic activity of ΔN-ClpC-F436A towards GFP-SsrA (Fig. 1D) and suggests more relaxed substrate selection and thus enhanced protein degradation upon NTD removal.

## Discussion

In the current work we show that the NTD functions as crucial regulatory element of ClpC activity control. Our findings explain how ClpC adaptors, pArg-substrates and toxic peptides lead to ClpC activation by binding to the NTD (Fig. 9). The NTD represses ClpC activity by contributing to formation of the inactive ClpC resting state. This is achieved by contacts of NTD residues to the AAA1 domain and the MD of the same and a neighboring subunit. Mutating the contact sites triggers ClpC activation by shifting the inactive resting state to a destabilized resting state or a functional hexamer. The MDs can still undergo head-to-head interactions in the activated NTD mutants, leading to the formation of active tetrahedral assemblies, but also more heterogenous complexes consisting of at least 4 hexamers. NTD residues involved in resting state formation are conserved in ClpC homologs (Appendix Fig. S10), indicating a preserved mode of ClpC activity control via the NTD, which we confirm for Mtb ClpC1. Notably, NTDs of one half-spiral do not contact the other half-spiral and interactions between the half-spirals are exclusively mediated by head-to-head MD contacts. This raises the question how NTD-AAA1 and

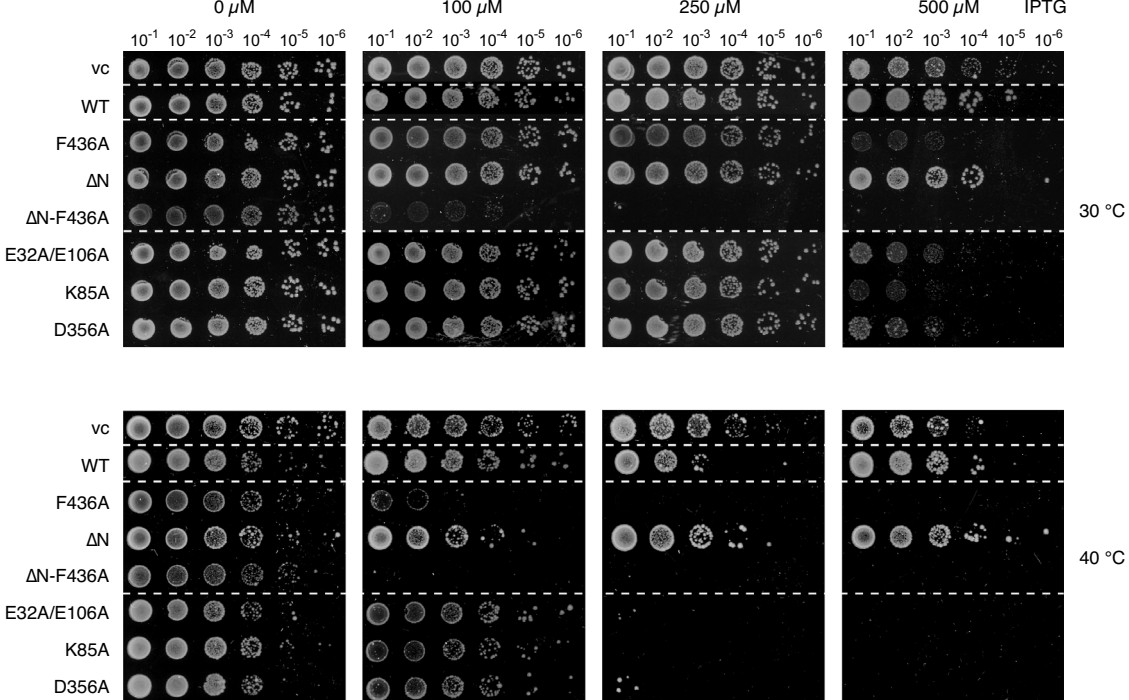

**Figure 8. Deregulated ClpC mutants create toxicity in vivo.**

*E. coli* cells expressing *S. aureus clpP* and harboring indicated plasmid-encoded *clpC* alleles under control of an IPTG-regulatable promoter were grown overnight at 30 °C and adjusted to an OD$_{600}$ of 1. Serial dilutions (10$^{-1}$–10$^{-6}$) were spotted on LB plates containing the indicated IPTG concentrations and incubated at 30 °C or 40 °C for 24 h. vc vector control. Source data are available online for this figure.

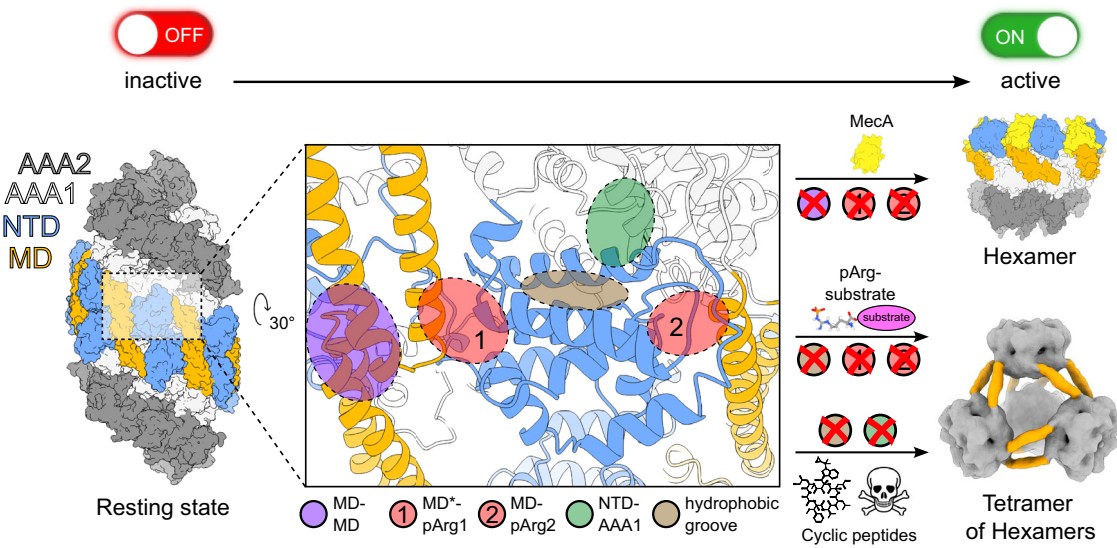

**Figure 9. Allosteric control of ClpC via its N-terminal domain.**

The structure of the ClpC resting state and crucial contact sites for resting state formation are indicated. The NTD interacts with the AAA1 domain of the same subunit and with M-domains of the same (MD) or a neighboring subunit (MD*) via its pArg1/2 binding sites. The NTD additionally harbors a hydrophobic groove for substrate binding. The adaptor MecA, pArg-substrates and cyclic peptides activate ClpC by binding to indicated sites and abrogating key interactions crucial for resting state formation. Binding of pArg-substrates or cyclic peptides creates hexamers that can interact via their MDs to form a tetrahedral assembly. ClpC/MecA complexes are hexameric as MecA binds to the tip of MDs blocking their interactions. Source data are available online for this figure.

NTD-MD interactions promote resting state formation. We suggest that NTDs are required for the helical arrangement of the AAA domains and accordingly NTD deletion in ΔN-ClpC leads to the formation of planar hexamers.

The NTD anchoring points include the pArg1/2 binding pockets, providing a rationale for ClpC activation by pArg-substrates (Morreale et al, 2022; Trentini et al, 2016). Binding of pArg will weaken NTD interactions in the resting state (Fig. EV2A), defining pArg as allosteric effector, which primes ClpC for activation. Similarly, mutating the pArg1/2 sites triggers ClpC activation. This effect can be either direct (T7D) or indirect (E32A, E106A) as these glutamate residues do not make contacts to the AAA1 or M-domains, implying that local structural alterations of the pArg1/2 sites lead to resting state destabilization.

Full activation of ClpC requires additional binding of another site of the substrate. pArg-substrates thus harbor two independent signals that can be physically separated: a phosphorylated arginine residue and an unfolded polypeptide stretch, represented in this study by FITC-casein (Fig. 9). We suggest that the NTD accommodates both entry portals for the two signals: the pArg binding pockets and a hydrophobic groove (Fig. EV2A), which was shown for the ClpC homologs ClpB and Hsp104 to interact with unfolded model proteins including casein (Rosenzweig et al, 2015). While the groove is not directly involved in NTD interactions within the resting state, we assume that substrate binding to this site causes further destabilization of the complex. The dual requirement for pArg-substrate induced ClpC activation strongly increases substrate specificity and prevents unwanted proteolysis. Our finding also explains the diverse efficiencies of ClpC/ClpP-dependent degradation of BacPROTAC model substrates in vitro, when most efficient degradation was noticed for a substrate harboring a fused pArg moiety and a long unstructured tail (Morreale et al, 2022).

How can NTDs sense pArg-substrates if both pArg1/2 binding sites are engaged in resting state interactions? In our cryo-EM structures we observed a high level of variability in ClpC stoichiometry and in flexibility of the NTDs (Fig. 5D; Movies EV2 and 3). NTDs of "back side" subunits of the resting state are mobile, raising the possibility that their pArg1/2-binding pockets become accessible for pArg-substrate binding. This might allow NTDs to sense the presence of pArg-substrates or adaptor proteins, causing mobilization of ClpC subunits from the resting state and ClpC activation. Alternatively, the generally increased dynamics of "back side" subunits facilitate their dissociation from the resting state. Accordingly, we show that ClpC can form resting states with diverse stoichiometries (10-mer to 14-mer). Dissociation will enable the released subunits to bind pArg-substrates or adaptor proteins, preventing their reassociation with the resting state and trigger ClpC activation. We therefore consider the diverse resting states to be physiologically relevant, yet ClpC resting state heterogeneity in vivo still needs to be documented. We infer that, in both scenarios, NTDs of "back side" subunits represent the activation region of the resting state.

All ClpC adaptor proteins characterized to date bind to the ClpC NTD, yet their interaction sites differ (Kirstein et al, 2007; Massoni et al, 2025; Wang et al, 2011). The adaptor protein MecA binds to both pArg1/2 binding pockets of the NTD and additionally to the tip of the M-domain (Wang et al, 2011), thus shielding all major sites involved in formation of the ClpC resting state (Figs. 9

and EV2B). The sporulation-specific MdfA adaptor binds to the hydrophobic groove of the NTD and additionally engages K85 through electrostatic interactions (Appendix Fig. S12A) (Massoni et al, 2025). Binding of MdfA to the NTD is incompatible with the ClpC resting state assembly (Appendix Fig. S12B). Together, these findings illuminate how adaptor protein binding will abrogate key NTD interactions within the resting state and trigger ClpC activation.

Our findings also explain how antibacterial peptides like cyclomarin A (CymA) cause Mtb ClpC1 deregulation. Activated ClpC NTD mutants form large tetrahedral assemblies as observed for CymA-activated ClpC1 (Maurer et al, 2019; Taylor et al, 2022), indicating a common pathway of ClpC/ClpC1 activation. The cyclic peptides bind to the hydrophobic groove of the NTD (Jagdev et al, 2023; Vasudevan et al, 2013; Wolf et al, 2019; Wolf et al, 2020) and therefore have been suggested to mimic substrate interactions (Hoi et al, 2023). While this model is appealing, the model substrate FITC-casein cannot autonomously activate ClpC, indicating that cyclic peptides must harbor additional features mediating ClpC activation. Here, we show that the NTD residue K85, which is in direct contact with all cyclic peptides bound to the Mtb ClpC1 NTD, is crucial for ClpC resting state formation (Jagdev et al, 2023; Vasudevan et al, 2013; Wolf et al, 2019; Wolf et al, 2020). Cyclic peptide binding will thus not only occupy the hydrophobic groove but will additionally abrogate a key interaction (K85-D364) necessary for ClpC1 resting state formation (Figs. 7A and EV1B). In support of this model, ClpC1-K85A is constitutively active (Fig. 7B,C). The dual feature of antibacterial peptides (binding to hydrophobic groove and K85) mechanistically explains how they trigger ClpC1 deregulation and cellular toxicity (Fig. 9). We conclude that antibacterial peptides hijack the NTD-dependent regulation of ClpC1 activity to cause lethal activation in combination with altered substrate specificity.

Next to its crucial function in controlling ClpC resting state formation we show that the NTD additionally downregulates ClpC activity in the context of single hexamers. Thus, ΔN-ClpC-F436A, which constitutively forms hexamers due to MD mutation, exhibits higher ATPase activity in presence of ClpP as compared to ClpC-F436A. ClpC-F436A requires substrate presence to reach similar ATP hydrolysis rates, indicating that substrate binding to the NTD overcomes repression. The NTD will thereby directly link substrate presence to full ClpC-F436A activation. Furthermore, ΔN-ClpC-F436A but not ClpC-F436A degrades GFP-SsrA, which is recognized by the central pore of Hsp100 proteins but not the NTD (Piszczek et al, 2005). This indicates that the NTD controls substrate flux to the central threading channel and increases substrate specificity. We suggest that the NTD additionally functions as selectivity barrier for substrates and removal of the NTD can lead to more unspecific protein degradation. Supporting such function, we noticed that expression of ΔN-ClpC-F436A in *E. coli* cells is much more toxic as compared to ClpC-F436A (Fig. 8). How exactly the NTD mediates its regulatory function in the context of a ClpC hexamer needs to be addressed in future studies.

In conclusion, we identify the NTD as central regulator of ClpC activity. Our findings illustrate how the NTD mechanistically links the availability of pArg-substrates and adaptor proteins to ClpC activation and how this regulatory function is hijacked by antibacterial cyclic peptides to deregulate ClpC activity.

# Methods

## Reagents and tools table

| Reagent/resource | Reference or source | Identifier or catalog number |
|---|---|---|
| **Experimental models** | | |
| *E. coli* XL1 blue | Agilent | 200249 |
| *E. coli* BL21 Rosetta DE3 | Novagen | 70954-3 |
| *E. coli* ΔclpB::kan | Katikaridis et al, 2021 | |
| *E. coli* ΔclpB | This study | |
| *E. coli* ΔclpP | Dougan et al, 2002 | |
| **Recombinant DNA** | | |
| Plasmids used in this study | See Appendix Table S2 | Appendix Table S2 |
| **Antibodies** | | |
| ClpC-specific antibody | Davids Biotechnologie | Self-made |
| anti-rabbit alkaline phosphatase conjugate | Vector Laboratories | AP-1000 |
| **Oligonucleotides and other sequence-based reagents** | | |
| pDS56 fw seq | Sigma-Aldrich | 5'-CTTTGTCTTCACCTCGAG-3' |
| SaClpC 400 bp seq | Sigma-Aldrich | 5'-CGTGCACAAGTTGTGAAAG-3' |
| SaClpC 1000 bp seq | Sigma-Aldrich | 5'-GTTTCCAACCTGTACAAG-3' |
| SaClpC 1600 bp seq | Sigma-Aldrich | 5'-CTAAACGACCAATTGGTAGC-3' |
| SaClpC 2000 bp seq | Sigma-Aldrich | 5'-CAACGATTTGCTGGATTC-3' |
| SaClpC E32A fw | Sigma-Aldrich | 5'-GGAACAGcACACCTATTATTGG-3' |
| SaClpC E32A rv | Sigma-Aldrich | 5'-CCAATAATAGGTGTgCTGTTCC-3' |
| SaClpC E106A fw | Sigma-Aldrich | 5'-GTTGGAACGGcACATATTTTATTAGGC-3' |
| SaClpC E106A rv | Sigma-Aldrich | 5'-GCCTAATAAAATATGTgCCGTTCCAAC-3' |
| SaClpC K85A fw | Sigma-Aldrich | 5'-[Phos]CACCTAGAGCTgcAAAAGTCATTGAATTATCG-3' |
| SaClpC K85A rv | Sigma-Aldrich | 5'-[Phos] TATAATGCAATGTACCAACATGATCTTG-3' |
| SaClpC D356A fw | Sigma-Aldrich | 5'-GGATTAAGAGcTCGTTACG-3' |
| SaClpC D356A rv | Sigma-Aldrich | 5'-CGTAACGAgCTCTTAATCC-3' |
| SaClpC R9A fw | Sigma-Aldrich | 5'-[Phos]GATTAACTGAGgcTGCACAGCGCGTATTAGC-3' |
| SaClpC R9A rv | Sigma-Aldrich | 5'-[Phos]TACCAAATAAGGATCCTCTCATAGTTAATTTCTCCTC-3' |
| SaClpC R122A fw | Sigma-Aldrich | 5'-[Phos]GTGTTGCAGCAgcAGTTTTGCAAATCTAGA-3' |
| SaClpC R122A rv | Sigma-Aldrich | 5'-[Phos]CTTCATTTTCACGAATCAAGCCTAATAAAATATGTTC-3' |
| SaClpC N462A fw | Sigma-Aldrich | 5'-[Phos]AATGAATGGAAGgcTGCACAAAATGGCATGT-3' |
| SaClpC N462A rv | Sigma-Aldrich | 5'-[Phos]TTTAGCTTCTTCATATTGCTTTTCAAGTTTTGTTTG-3' |
| SaClpC T7D fw | Sigma-Aldrich | 5'-GGTAGATTAgaTGAGCGTGCAC-3' |
| SaClpC T7D rv | Sigma-Aldrich | 5'-GTGCACGCTCAtcTAATCTACC-3' |
| MtClpC K85A fw | Sigma-Aldrich | 5'-[Phos]TACCCCCCGCGCCgcAAAGGTCCTCGAG-3' |
| MtClpC K85A rv | Sigma-Aldrich | 5'-[Phos]AACGGAATGTGCCCAGACGGCGCCTGCTG-3' |
| MtClpC D364A fw | Sigma-Aldrich | 5'-[Phos]TCAAGGGCCTGCGGGcCCGGTACGAGGC-3' |
| MtClpC D364A rv | Sigma-Aldrich | 5'-[Phos]GGATCTCGATGGTGTGCTCCACCGTCGGCTCACC-3' |
| **Chemicals, enzymes and other reagents** | | |
| Kanamycin | Sigma | K4000 |
| Ampicillin | Roth | K029.2 |
| Protino™ Ni-IDA | Macherey-Nagel | MN745210.30 |
| Protease inhibitor Pepstatin A | Roth | 2936.3 |

| Reagent/resource | Reference or source | Identifier or catalog number |
|---|---|---|
| Protease inhibitor Aprotinin | Roth | A162.3 |
| Protease inhibitor Leupeptin | Roth | CN33.4 |
| DNaseI | Roche | 10104159001 |
| Phusion HF DNA-Polymerase | ThermoScientific | F-530 |
| T4 DNA Ligase | NEB | M0202L |
| Imidazole | Roth | 3899.4 |
| Pyruvate kinase | Sigma | P7768 |
| FITC-casein (Casein fluorescein isothiocyanate) | Sigma | C0528 |
| Bradford reagent | BioRad | 5000006 |
| Phospho-L-Arginine (lithium salt hydrate) | Cayman Chemical | 23123 |
| Arg | Sigma | A8094 |
| ATP | Sigma | A26209 |
| NADH | Sigma | 43420 |
| PK/LDH | Sigma-Aldrich | P0294 |
| PEP (Phospho(enol)pyruvic acid trisodium salt hydrate) | Sigma | P7002 |
| ATPγS | Roche | 11162306001 |
| Glutaraldehyde | Sigma | G6403 |
| SYPRO Ruby Protein Gel Stain | Thermo Fisher | S12000 |
| Uranyl acetate | Ted Pella | 19481 |
| IPTG (Isopropyl-β-D-1-thiogalactopyranoside) | Roth | CN08.3 |
| BSA | Serva | 11930.04 |
| ECF substrate | GE Healthcare | RPN3685 |
| **Software** | | |
| FIJI | https://imagej.net/software/fiji/downloads | |
| Prism 10 | Graphpad | |
| ChimeraX 1.8 | Pettersen et al, 2021 | |
| Prometheus Panta Analysis software (version 1.4.3) | NanoTemper | |
| EPU software | ThermoFisher | |
| EMAN2 v.2.12 | Tang et al, 2007 | |
| IMAGIC-4D | van Heel et al, 1996 | |
| cryoSPARC v4.7.1 | Huber et al, 2018 | |
| Clustal Omega | https://www.ebi.ac.uk/Tools/msa/clustalo/ | |
| Jalview | Waterhouse et al, 2009 | |
| Refeyn DiscoverMP software | Refeyn TwoMP | |
| AlphaFold 3 | Jumper et al, 2021; Varadi et al, 2024 | |
| AlphaFill | Hekkelman et al, 2023 | |
| Phenix v.1.21.2 | Adams et al, 2010 | |
| Coot v.0.9 | Emsley et al, 2010 | |
| **Other** | | |
| French Pressure cell press | Sim-Aminco | |
| Superdex S200 | Cytiva | 28-9893-35 |
| CLARIOstar^plus plate reader | BMG Labtech | |
| Prometheus™ Panta | NanoTemper | |
| Prometheus™ standard capillaries | NanoTemper | PR-C002 |
| ÄKTA pure | Cytiva | |

| Reagent/resource | Reference or source | Identifier or catalog number |
|---|---|---|
| Superose 6 Increase 10/ 300 GL column | Cytiva | 29-0915-96 |
| Talos L120C electron microscope | Thermo Fisher Scientific | |
| Image-Reader LAS-4000 | Fujifilm | |
| Refeyn Mass Photometer | Refeyn | |
| GloQube | Quorum | |
| Vitrobot Mark IV | Thermo Fisher Scientific | |
| Talos Arctica | Thermo Fisher Scientific | |
| 300-kV Titan Krios G2 microscope | Thermo Fisher Scientific | |
| 96-well plates (black) | Greiner | 655079 |
| 96-well plates (transparent) | Greiner | 655180 |

## Strains and plasmids

All strains and plasmids used in this study are summarized in the Appendix Table S2. *Escherichia coli* (*E. coli*) cells were grown in LB medium at 30 °C containing appropriate antibiotics with agitating speed 120 rpm. For protein overproduction 2x YT medium (16 g/l tryptone, 10 g/l yeast extract, 5 g/l NaCl, adjusted to pH 7) was used. *E. coli* XL1 blue was used for cloning and retaining of plasmids requiring kanamycin at 50 µg/ml and ampicillin at 100 µg/ml for plasmid propagation.

## Protein purification

*Staphylococcus aureus* (*Sa*) ClpC (WT and derivatives), ClpP and MecA were purified after overproduction from *E. coli* Δ*clpB::kan* cells using pDS56-derived expression vectors (Carroni et al, 2017). *Mycobacterium tuberculosis* (*Mtb*) ClpC1 (wild-type and mutant derivatives) was purified after overproduction in *E. coli* BL21 cells using pET24a-derived expression vectors. GFP-SsrA was purified after production in *E. coli* Δ*clpP* cells (Dougan et al, 2002). ClpC deletion mutants were generated by PCR and point mutants were constructed by Quikchange one-step site-directed mutagenesis. All mutations were verified by sequencing. ClpC, ClpP and MecA harbor a C-terminal His6-tag while GFP-SsrA harbors an N-terminal His6-tag and were purified using Ni-IDA (Macherey-Nagel) following the instructions provided by the manufacturer. In short, cell pellets were resuspended in buffer A (50 mM NaH$_2$PO$_4$, 300 mM NaCl, 5 mM β-mercaptoethanol, pH 8.0) supplemented with protease inhibitors (Roche). After cell lysis using a French press the cell debris was removed by centrifugation at 18,000 × *g* for 60 min at 4 °C and the cleared lysate subsequently transferred into a plastic column containing pre-equilibrated 0.8–1 g Protino™ Ni-IDA resin (Macherey-Nagel) at 4 °C. Afterwards the resin was washed once with buffer A. His-tagged proteins were eluted by addition of buffer A supplemented with 250 mM Imidazole. Subsequently, pooled protein fractions were subjected to size-exclusion chromatography (Superdex S200, Amersham) in buffer B (50 mM Tris-HCl pH 7.5, 50 mM KCl, 20 mM MgCl$_2$, 5% (v/v) glycerol, 2 mM DTT).

Pyruvate kinase of rabbit muscle and Casein fluorescein isothiocyanate (FITC-casein) from bovine milk were purchased from Sigma. Protein concentrations were determined with the Bradford assay (BioRad).

## Biochemical assays

Buffer C (50 mM Tris-HCl pH 7.5, 50 mM KCl, 20 mM MgCl$_2$, 2 mM DTT) was used for all biochemical assays if not mentioned otherwise. Assays in presence of pArg were conducted in buffer D (50 mM Tris-HCl pH 7.5, 25 mM KCl, 10 mM MgCl$_2$, 2 mM DTT).

## ATPase assay

Rates of ATP hydrolysis were determined by a coupled-colorimetric assay as described before (Oguchi et al, 2012). Oxidation of NADH to NAD$^+$ was monitored by measuring the absorbance at a wavelength of 340 nm.

ATPase activity measurements were performed at final concentrations of 0.5 µM ClpC, 0.75 µM ClpP, 0.75 µM MecA, 0.5 µM FITC-casein and an ATP regenerating system (PK/LDH mix: 250 µM NADH, 500 µM PEP, 1/20 (v/v) PK/LDH (Sigma-Aldrich)) in buffer C/D supplemented with 2 mM ATP at 30 °C in a transparent 96-well plate (Greiner) using a CLARIOstar$^{plus}$ plate reader (BMG Labtech). In the case of samples containing pArg, 250 µM Phospho-L-Arginine (lithium salt hydrate, Cayman Chemical) was added and buffer D was used.

The ATP hydrolysis rate for at least three biological replicates was calculated according to the following equation:

$$\text{ATPase rate} = \frac{1}{\varepsilon_{\text{NADH}} \times C_{\text{ClpC}} \times d} \times \frac{\Delta A_{340\text{nm}}}{\Delta t}$$

$\varepsilon_{\text{NADH}}$ – molar absorption coefficient of NADH at a wavelength of 340 nm (6 220 M$^{-1}$ cm$^{-1}$).
$C_{\text{ClpC}}$ – final concentration of ClpC (0.5 µM).
d – optical path length (1 cm).
$\Delta A_{340\,\text{nm}}/\Delta t$ – slope of the linear decline in absorption at a wavelength of 340 nm.

## Degradation assays

Proteolytic activity was determined using fluorescein labelled casein (FITC-casein). The increase of fluorescence upon FITC-casein degradation was monitored at 483-14 nm and 530-30 nm as excitation and emission wavelengths, respectively, in black 96-well plates (Greiner) using a CLARIOstar$^{plus}$ plate reader (BMG Labtech).

FITC-casein degradation measurements were performed at final concentrations of 0.5 µM ClpC, 0.75 µM ClpP, 0.75 µM MecA, 0.5 µM FITC-casein and an ATP regenerating system (3 mM PEP, 20 ng/µl PK (Sigma-Aldrich)) in buffer C/D. In the case of samples containing pArg, 250 µM Phospho-L-Arginine (lithium salt hydrate, Cayman Chemical) was added and buffer D used. The raw fluorescence signal was normalized by setting the initial value to 100%. The FITC-casein fluorescence gain (% min$^{-1}$) was calculated by determining the initial slopes of the linear fluorescence signal increase for at least three biological replicates.

Degradation of GFP-SsrA was performed in buffer D using the following final protein concentrations: 0.5 µM ClpC, 1 µM ClpP, 0.75 µM MecA, 0.5 µM GFP-SsrA and an ATP regenerating system (3 mM PEP,

20 ng/µl PK (Sigma-Aldrich)). GFP fluorescence was monitored at 470-15 nm and 515–20 nm as excitation and emission wavelengths, respectively, in black 96-well plates (Greiner) using a CLARIOstar^plus plate reader (BMG Labtech). The degradation rate (% min^−1) was calculated by determining the initial slopes of the linear fluorescence signal increase for at least three biological replicates. Initial slopes of control reactions including only ClpP were substracted.

## Anisotropy measurements

FITC-casein binding was monitored by anisotropy measurements. 100 nM FITC-casein was pre-incubated with varying concentrations of ClpC (0.04-10 µM) in buffer D in presence of 2 mM ATPγS at 25 °C for 10 min. Changes in fluorescence polarization were determined with a CLARIOstar^plus plate reader (BMG Labtech) by measuring the polarization at 482-16 nm and 530-40 nm as excitation and emission wavelengths, respectively. Ten technical and two biological replicates were performed for each sample.

## Glutaraldehyde crosslinking

For Glutaraldehyde (GA) crosslinking samples were pre-incubated with 2 mM ATPγS at 25 °C for 10 min. The following concentrations were used: 1 µM ClpC WT in absence and presence of 1.5 µM MecA or 1 µM ClpC-F436A, ClpC-ΔN, ClpC-ΔN-F436A in buffer E (50 mM HEPES pH 7.5, 25 mM KCl, 10 mM MgCl₂, 2 mM DTT). Crosslinking was initiated by addition of GA (Sigma) to a final concentration of 0.1%. Samples were taken after 0/2/10 min (timepoint 0 was taken prior GA addition) and crosslinking was quenched by adding Tris (pH 7.5) to a final concentration of 50 mM. Additionally, SDS sample buffer was added and after boiling for 5 min at 95 °C samples were subjected to SDS-PAGE analysis. Gels were stained with SYPRO Ruby Protein Gel Stain (Thermo Scientific).

## Dynamic light scattering (DLS)

Dynamic light scattering (DLS) was performed with a Prometheus™ Panta (NanoTemper) at 25 °C using Prometheus™ standard capillaries (NanoTemper). The hydrodynamic radius of ClpC complexes in absence and presence of MecA/ClpP/FITC-casein was determined using 4 µM ClpC or respective variants, 6 µM MecA, 6 µM ClpP and 4 µM FITC-casein in buffer C/D without crosslinking reagents. Samples were pre-incubated for 5–10 min at RT with 2 mM ATPγS, in case of samples containing pArg, 500 µM Phospho-L-Arginine (lithium salt hydrate, Cayman Chemical) was added additionally. Three technical of two biological replicates were taken for each sample. Single capillaries were scanned 10 times. Hydrodynamic radii were determined from particle size distribution peak values using the Prometheus Panta Analysis software (version 1.4.3). Statistical significance was tested using Brown–Forsythe and Welch one-way ANOVA tests (Unpaired $t$ with Welch's correction).

## Analytical size-exclusion chromatography (SEC)

Complex formation of ClpC and respective variants was monitored via analytical size-exclusion chromatography. SEC was performed at 4 °C using an ÄKTA pure (Cytiva) equipped with a Superose 6 Increase 10/300 GL column (Cytiva), pre-equilibrated with buffer B supplemented with 2 mM ATP. 6 µM ClpC-E280A/E618A (in presence and absence of 9 µM ClpP and/or 6 µM FITC-casein) was pre-incubated with 2 mM ATPγS at RT for 5 min and 200 µl of the sample was loaded onto the column. Aliquots were taken and subjected to SDS-PAGE analysis. Gels were stained using SYPRO Ruby Protein Gel Stain. For each sample, two biological replicates were performed.

FITC-casein elution profiles were monitored by fluorescence endpoint measurements using a CLARIOstar^plus plate reader (BMG Labtech) with the previously described filter settings.

## Negative stain electron microscopy

Negative staining, data collection, and processing were performed as described previously (Gasse et al, 2015). In brief, 5 µl sample were applied to a glow-discharged grid with an 6–8 nm thick layer of continuous carbon. After incubation for 5–60 s the sample was blotted on a Whatman filter paper 50 (1450-070) and quickly washed with three drops of water. Samples on grids were stained with 2.5% aqueous uranyl acetate. Images were acquired using a ThermoFisher Talos L120C electron microscope equipped with a Ceta 16 M camera, operated at 120 kV. The micrographs were acquired at ×57,000 magnification (resulting in 2.26 Å per pixel) using EPU software. For 2D classification 10-25k particles were selected using the boxing tool in EMAN2 (box pixel sizes 200 (ClpC-DWB, ClpC-F436A-DWB + ClpP, ΔN-ClpC-F436A + ClpP), 208 (ClpC-WT, ClpC-WT + pArg, ClpC-E32A/E106A, ClpC-D356A), or 300 (ClpC-WT + FITC-casein + pArg, ClpC-E32A/E106A + FITC-casein, ClpC-D356A-DWB + FITC-casein) (Tang et al, 2007). Particles forming rod-like structures consisting of ΔN-ClpC-DWB + ClpP were picked via the cryoSPARC filament tracer (Huber et al, 2018) with a box pixel size of 320. Image processing was carried out using the IMAGIC-4D package (van Heel et al, 1996). Particles were band-pass filtered, normalized in their gray value distribution, and mass centered. 2D alignment, classification, and iterative refinement of class averages were performed as previously described (Liu and Wang, 2011).

To obtain 3D volumes of the ClpC-WT and the ClpC-WT + FITC-casein + pArg samples, cryoSPARC v4.7.1 (Huber et al, 2018) was used. For the ClpC-WT sample, 230 particles were first picked manually, classified into 3 classes and used as template for picking from 421 micrographs. A total of 93.045 particles were picked and upon 2D classification and selection, 19.055 particles were extracted. Ab initio reconstructions were performed with single and multiple, up to 3, starting models. All models obtained could be refined to a consensus map of the resting state, at around 11 Å resolution. For the ClpC-WT + FITC-casein + pArg sample, using the blob picker (200–250 Å diameter), 342.586 particles were picked and 2D classified. Particles from noisy classes were removed and, upon visual inspection, two groups of particles were obtained: tetrahedron arrangements and dimers of hexamers. For the dimer (20.077 particles) 3D ab initio reconstruction, followed by homogeneous or non-uniform refinements, was performed to obtain a dimer of planar hexamers placed in a head-to-head arrangement at approx. 12 Å resolution.

ClpC and analyzed mutants were pre-incubated without crosslinking reagents for 15 min in the presence of 2 mM ATPγS at RT and diluted rapidly before application on the grid to the final concentration. The following concentrations were used: 4 µM ClpC WT + 4 µM FITC-casein + 500 mM pArg (diluted 10-fold),

4 μM ClpC-E32A/E106A ± 4 μM FITC-casein (diluted 20-fold), 1 μM ClpC-D356A (diluted 2-fold). To assess complex formation of ClpC-D356A with FITC-casein, a ClpC-D356A-DWB variant in presence of FITC-casein was subjected to analytical SEC as described previously, with the peak fraction of the early eluting large complexes was applied to a grid.

## In vivo toxicity assays (Spot tests)

*E. coli* cells harboring plasmids coding for *S. aureus* ClpP and ClpC-WT and ClpC variants under an IPTG-controlled promoter were grown at 30 °C, 120 rpm overnight. Serial dilutions ($10^{-1}$ to $10^{-6}$) after adjustment to $OD_{600} = 1$ were spotted on LB agar plates supplemented with different IPTG concentrations (0, 100, 250, 500 μM IPTG) using a custom-made spotter (ZMBH workshop, University of Heidelberg). Plates were incubated overnight at 30/37/40 °C. Two biological replicates were performed.

## Western blotting

Total cell extracts were prepared and separated via SDS-PAGE and subsequently electrotransferred onto a polyvinylidene fluoride (PVDF) membrane. Next, the membrane was incubated in blocking solution (3% bovine serum albumin (w/v) in TBS) for at least 30 min at RT. Protein levels were determined by incubating the membrane with ClpC-specific antibodies (1:50,000 in TBS-T + 3% (w/v) bovine serum albumin) and an anti-rabbit alkaline phosphatase conjugate (Vector Laboratories, AP1000) as secondary antibody (1:20,000). ClpC-specific antibodies were produced in rabbit (https://davids-bio.com). Blots were developed using ECF Substrate (GE Healthcare) as reagent and imaged using an Image-Reader LAS-4000 (Fujifilm).

## Bioinformatic analyses

Multiple sequence alignments were performed using Clustal Omega (https://www.ebi.ac.uk/Tools/msa/clustalo/) and displayed using Jalview (Waterhouse et al, 2009).

## Mass photometry

*S. aureus* ClpC WT was incubated at 20 μM concentration in a buffer containing 25 mM Tris-HCl (pH 7.5), 25 mM KCl, 10 mM $MgCl_2$, 1 mM DTT and 2 mM ATP for 10 min at 30 °C for resting state formation. After incubation, the sample was diluted to achieve desired protein concentrations (50 nM, 100 nM, 500 nM or 1 μM) directly on the Refeyn Mass Photometer gasket. Recordings of the sample were performed right after dilution. Calibrating the curves and plotting of the measurements was done using the Refeyn DiscoverMP software.

## Cryo-EM sample vitrification and data collection

*S. aureus* ClpC WT at 10 μM concentration was incubated for 10 min at room temperature in a buffer containing 25 mM Tris-HCl (pH 7.5), 25 mM KCl, 10 mM $MgCl_2$, 1 mM DTT, along with 2 mM ATP for resting state formation. After incubation, 3 μL of oligomerized *S. aureus* ClpC WT was applied to glow-discharged Ultrafoil R2/2 (Au 200 mesh) grids, which were treated for 60 s at 40 mA using a GloQube (Quorum) instrument. The grids were then plunged in liquid ethane for vitrification using a Vitrobot Mark IV (Thermo Fisher Scientific) set at 100% humidity and 24 °C, with a blot force of 1 and a blotting time of 2 s. After vitrification, the grids were clipped and screened using a Talos Arctica microscope (Thermo Fisher Scientific) to assess the presence of oligomeric ClpC WT, ice thickness, and particle distribution in ice. Data was collected from the best-looking grid using a 300-kV Titan Krios G2 microscope (Thermo Fisher Scientific) equipped with Bioquantum energy filter and a K3 direct electron detector (Gatan/Ametek). A total of 15,057 micrographs were acquired at 105,000x nominal magnification, corresponding to a pixel size of 0.828 Å, with a total exposure dose of 40 e/Å² and defocus values ranging from −2.2 to −1.8 μm with a step of 0.2 μm.

## Cryo-EM data processing

Recorded movies were pre-processed using CryoSPARC Live 4.6.0 to perform motion correction and CTF estimation (Appendix Fig. S6) (Punjani et al, 2017). Initially, a blob picker was used with a particle diameter range of 140–260 Å and a box size of 420 pixels, which was subsequently binned down to 92 pixels for fast 2D classification. Selected 2D classes were used as templates for template-based particle picking. The resulting 1,301,151 particles were re-extracted using a box size of 512 pixels, subsequently binned down to 256 pixels. Multiple rounds of 2D classification and ab-initio volumes generation facilitated classifying out noisy particles (junk), generating volumes of the 10-mer, 12-mer, and 14-mer (Punjani et al, 2017). A total of 693,338 particles were finally selected for 3D classification with alignment (heterogeneous refinement), utilizing ab-initio volumes of the 10-mer, 12-mer, and 14-mer. Each 3D class was refined using homogeneous and non-uniform (NU) refinement. Further refinement of each oligomeric resting state (10-mer, 12-mer and 14-mer) were carried out after particles unbinning to increase resolution and obtain consensus maps. Additionally, the unsharpened 14-mer was sharpened using LAFTER (Punjani et al, 2017; Punjani et al, 2020; Ramlaul et al, 2019).

The 14-mer map underwent further focused refinement after signal subtraction of each dimeric unit in the resting state (Appendix Fig. S6). A volume was generated around a dimeric unit using ChimeraX 1.8 (Pettersen et al, 2021). The volume was imported into cryoSPARC and converted into a mask and an inverted mask using the volume tools function with a dilation radius of 1 and soft padding of 4. The inverted mask was used to perform signal subtraction on the 14-mer particles. The signal-subtracted particles were locally refined using the mask, a process applied to each of the dimeric units. Using their half maps, all seven locally refined dimeric units were combined into a single map using the PHENIX *combine-focus-map tool* (Adams et al, 2010). Local resolution was estimated using PHENIX *Local Resolution Estimation*, estimating the mean resolution of the combined focus map at 3.2 Å. Resolution was also estimated using the EMDB Fourier Shell Correlation Server (https://www.ebi.ac.uk/emdb/validation/fsc/results/), determining the resolution at 3.6 Å.

Further data processing was conducted to analyze dynamics of the 14-mer volume seam region. A mask was generated around the two monomers forming the seam region, which was used to perform 3D classification without alignment and 3D variability analysis in cryoSPARC. The 3D classification was performed at

10 Å resolution with 10 classes. The 3D Variability Analysis was also performed at 10 Å resolution and the 3D Variability Display was run in cluster output mode (Punjani and Fleet, 2021). The cryoDRGN algorithm was used to investigate NTD motion in the 14-mer (Zhong et al, 2021). To do so, the cryoDRGN algorithm was trained on the particles constituting the 14-mer volume with a dimension of 8 for latent variables, 50 training epochs, 256 encoder and decoder layers. These parameters were chosen according to the cryoDRGN tutorial (Zhong et al, 2021) (https://ez-lab.gitbook.io/cryodrgn/cryodrgn-empiar-10076-tutorial). Analysis of the training was performed using the cryodrgn analyse tool from cryoDRGN which generated volumes that were displayed and morphed together using UCSF ChimeraX.

## Model building

The initial monomeric atomic model of *S. aureus* ClpC WT was predicted through AlphaFold 3 (Jumper et al, 2021; Varadi et al, 2024). Each domain of the predicted model was rigidly fitted into the 14-mer consensus map using UCSF ChimeraX, resulting in a fitted monomeric atomic model. The geometry of the fitted monomeric atomic model was improved using PHENIX *Geometry Minimization* and further refined against the ClpC 14-mer EMready sharpened map using PHENIX *Real Space Refinement*.

Once ideal scores were attained, seven refined monomeric atomic models were rigidly fitted using ChimeraX to generate an initial 7-mer model for determining the non-crystallographic symmetry (NCS) operators using *Find NCS* in Phenix. The NCS operators were then applied to one refined monomeric atomic model using *Apply NCS Operators* in Phenix, followed *by Real Space Refinement* of the 7-mer model against the EMready sharpened 14-mer map. To complete the model, a copy of the 7-mer model was rigidly fitted into the other half of the 14-mer map and refined against the ClpC 14-mer sharpened maps using *Real Space Refinement*. Several iterations between *Real Space Refinement* and manual atomic refinement in Coot were performed on the 14-mer model until its geometry and fit could not be further improved. Unaccounted density within the 14-mer, along with nucleotide docking using AlphaFill, helped identify nucleotide-binding regions within the 14-mer atomic model (Hekkelman et al, 2023). When the density was clear and identifiable, ATP or ADP was built and fitted into the density using *Get Monomer* and *Ligand Jiggle Fit* in Coot (Emsley et al, 2010). Further manual refinements in Coot and *Real Space Refinement* were performed to ensure proper fitting of the ligands. A final round of Real Space Refinement was conducted with the 14-mer and nucleotides model against the raw 14-mer map.

The 14-mer model was used as a starting model for the 10- and 12-mer. Initial atomic models of 10- and 12-mer were generated by removing dimeric units from the 14-mer model. Each model was refined against their corresponding raw map using *Real Space Refinement*. Some iterations between manual model refinement in Coot and *Real Space Refinement* using the Cryosparc sharpened map occurred to improve model fit and geometry. Once achieved, nucleotide states established and built in the model as described previously. A final round of *Real Space Refinement* was conducted with the model and nucleotides against the raw maps.

## Data availability

All data are contained within the manuscript. All cryo-EM density maps, half maps, masks, Fourier shell correlation curves and composite maps were deposited into the Electron Microscopy Data Bank (https://www.ebi.ac.uk/pdbe/emdb/) under accession codes EMD-53014 for the 14-mer, EMD-53324 for the 12-mer and EMD-53312 for the 10-mer. The corresponding model coordinates were deposited in the Protein Data Bank (https://www.ebi.ac.uk/pdbe) under accession codes 9QCL, 9QRW and 9QQR. Structural maps are accessible via https://oc.embl.de/index.php/s/qNNhX08oVWeso8h.

The source data of this paper are collected in the following database record: biostudies:S-SCDT-10_1038-S44318-025-00575-1.

## Peer review information

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

## Acknowledgements

TJ was supported by the Heidelberg Biosciences International Graduate School (HBIGS). The cryo-EM data was collected at the Cryo-EM Swedish National Facility funded by the Knut and Alice Wallenberg, Family Erling Persson and Kempe Foundations, SciLifeLab, Stockholm University and Umeå University. We thank the facility personnel Dustin Morado, Karin Walldén and Mathieu Coinçon. We would like to acknowledge access to the infrastructure and support provided by the Cryo-EM Network at the Heidelberg University (HDcryoNet). This work was supported by a grant of the Deutsche Forschungsgemeinschaft to AM (MO970/8-1). LE and SA are supported by the Knut and Wallenberg Foundation grant 2021.0347 and by Stockholm University. MC is supported supported by SciLifeLab as part of the Cryo-EM National Infrastructure Unit and by the Stiftelsen för Strategisk Forskning (SSF) grant RIF-21.

## Author contributions

**Timo Jenne**: Conceptualization; Resources; Formal analysis; Supervision; Investigation; Visualization; Methodology; Writing—review and editing. **Lisa Engelhardt**: Conceptualization; Formal analysis; Investigation; Visualization; Methodology; Writing—review and editing. **Ieva Baronaite**: Formal analysis; Investigation; Visualization; Writing—review and editing. **Dorit Levy**: Formal analysis; Investigation; Writing—review and editing. **Inbal Riven**: Writing—review and editing. **Maciej Malolepszy**: Formal analysis; Investigation. **Stavros Azinas**: Formal analysis. **Taras Sych**: Resources; Formal analysis. **Erdinc Sezgin**: Resources. **Dirk Flemming**: Resources; Software. **Irmgard Sinning**: Resources. **Gilad Haran**: Writing—review and editing. **Marta Carroni**: Conceptualization; Resources; Formal analysis; Supervision; Funding acquisition; Investigation; Visualization; Methodology; Writing—review and editing. **Axel Mogk**: Conceptualization; Resources; Formal analysis; Supervision; Funding acquisition; Investigation; Visualization; Methodology; Writing—original draft.

Source data underlying figure panels in this paper may have individual authorship assigned. Where available, figure panel/source data authorship is listed in the following database record: biostudies:S-SCDT-10_1038-S44318-025-00575-1.

## Funding

## Disclosure and competing interests statement

The authors declare no competing interests.

# Expanded View Figures

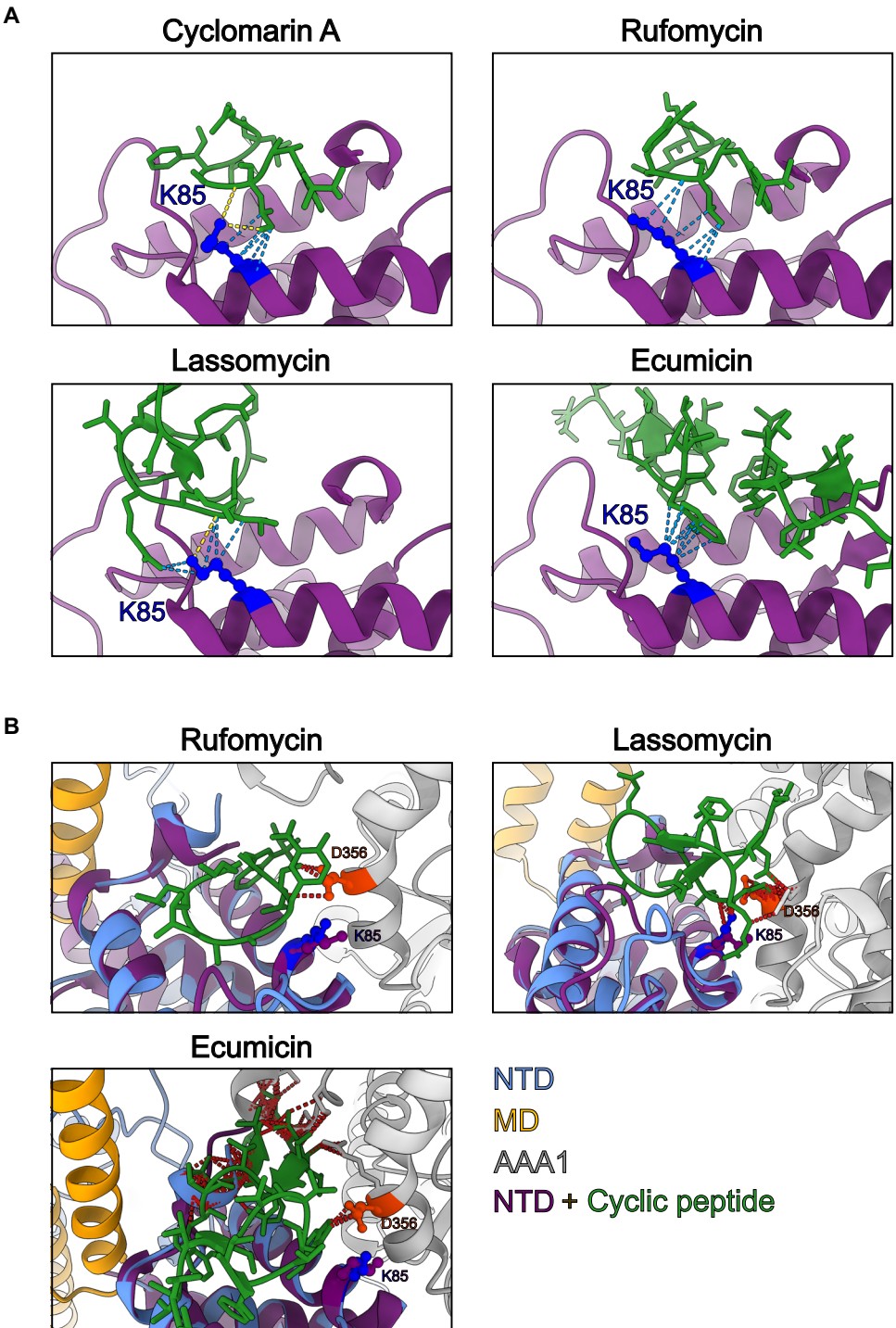

**Figure EV1. Antibacterial peptides abrogate key NTD-AAA1 interaction.**

(A) Co-crystal structures (pdbs: 3wdc, 6cn8, 8ibp, 6pbs) of indicated cyclic peptides (green) bound to the Mtb ClpC1 NTD (purple) are displayed. The NTD residue K85 is highlighted in blue and interactions with the respective cyclic peptides are indicated (H-bonds in yellow, contacts in light blue). (B) Co-crystal structures of Mtb ClpC1 and cyclic peptides were superimposed with an NTD of the *S. aureus* ClpC resting state. Cyclic peptides clash with the conserved AAA1 residue D356 (D364 in Mtb ClpC1), indicated with dark red dashed lines, abrogating the crucial NTD-AAA1 interaction.

**A**

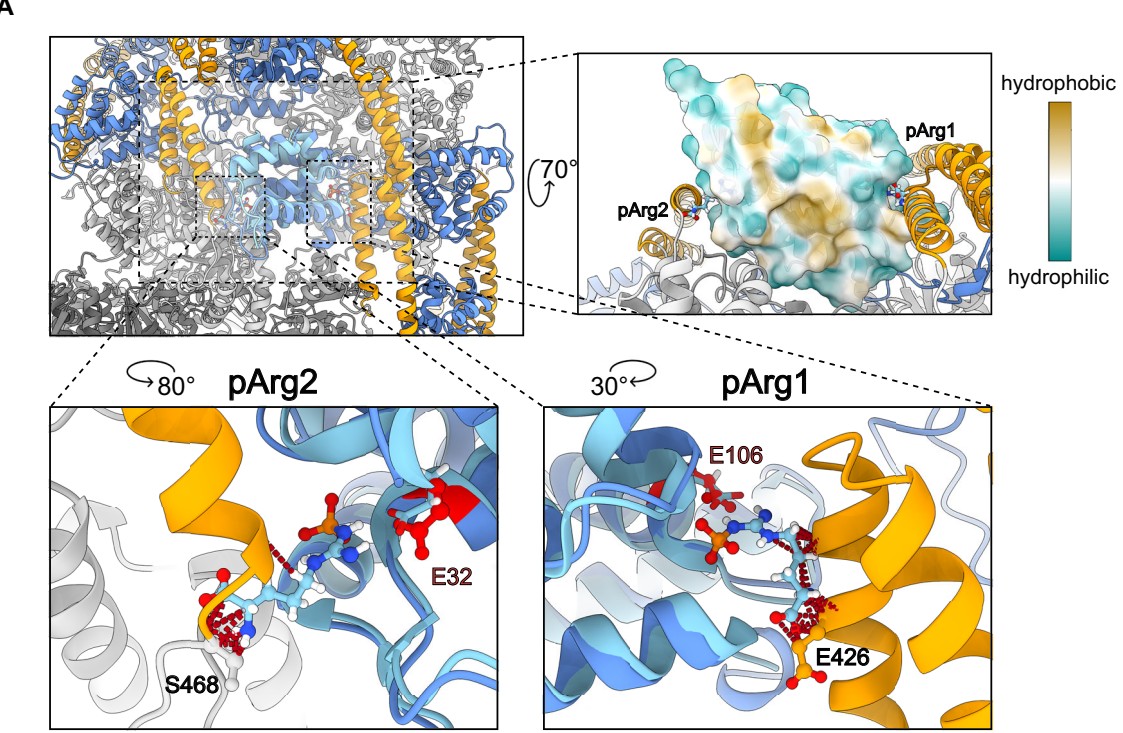

**B**

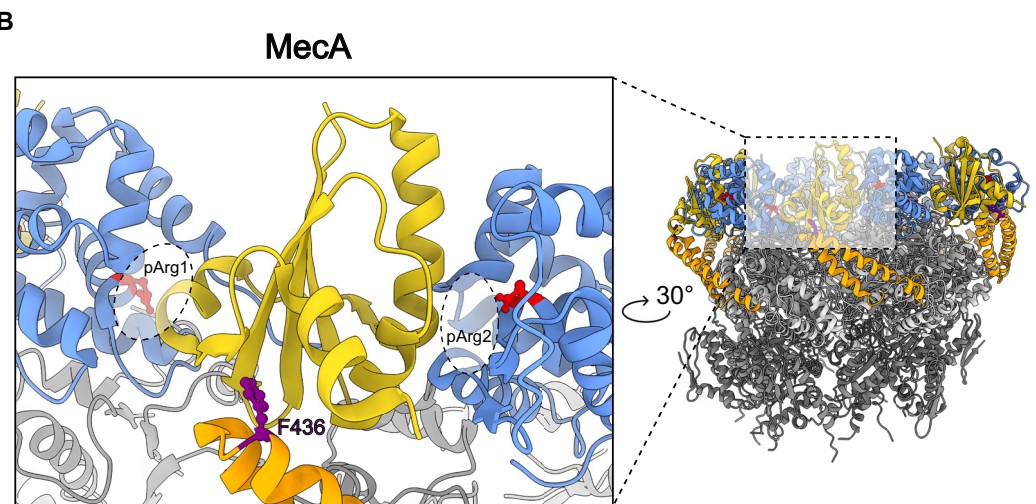

**Figure EV2.  Activation of ClpC by binding of pArg and MecA.**

(**A**) The *B. subtilis* NTD (cyan) co-crystallized with pArg (pdb:5hbn) was superimposed with an NTD of the resting state. pArg1/2 binding sites are involved in resting state formation by contacting MDs of the same (pArg2) and a neighboring subunit (pArg1). pArg binding will abrogate these interactions leading to resting state destabilization (clashes of pArg with the MD are indicated with dark red dashed lines). Additional binding of a disordered segment of a pArg-substrate to the hydrophobic groove of the NTD is required for ClpC activation. (**B**) MecA binds to both pArg1/2 binding sites and the MD tip including the conserved residue F436, thereby shielding major sites involved in resting state formation (pdb:3j3s).

