## [Peer Review File · The EMBO Journal]

Allosteric control of the bacterial ClpC/ClpP protease and its hijacking by antibacterial peptides

Timo Jenne, Lisa Engelhardt, Ieva Baronaite, Dorit Levy, Inbal Riven, Maciej Malolepszy, Stavros Azinas, Tara Sych, Erdinc Sezgin, Dirk Flemming, Irmgard Sinning, Gilad Haran, Marta Carroni, and Axel Mogk

Corresponding author(s): Axel Mogk (a.mogk@zmbh.uni-heidelberg.de), Marta Carroni (marta.carroni@scilifelab.se)

Review Timeline:

Submission Date:	24th Mar 25
Editorial Decision:	13th May 25
Revision Received:	21st Jul 25
Editorial Decision:	14th Aug 25
Revision Received:	19th Aug 25
Accepted:	7th Sep 25

Editor: Cornelius Schneider

Transaction Report:

Dear Dr. Mogk,

Thank you for submitting your manuscript for consideration by the EMBO Journal. It has now been seen by two referees whose comments are shown below. We were not able to obtain the feedback of the third referee until now and we have therefore decided to not wait any further and proceed with these two reports. We will share the missing report with you in case we still receive the feedback but will only take it into consideration for our assessment if it voices critical technical concerns.

As you can see from the reports both referees think that the presented data are interesting and that the experimental quality is high. Given the referees' positive recommendations, I would like to invite you to submit a revised version of the manuscript, addressing the comments of all three reviewers. I should add that it is EMBO Journal policy to allow only a single round of revision, and acceptance of your manuscript will therefore depend on the completeness of your responses in this revised version. I am happy to discuss questions regarding possible revisions via e-mail or videoconferencing.

Thank you for the opportunity to consider your work for publication. I look forward to your revision.

Yours sincerely,

Cornelius Schneider, PhD
Editor
The EMBO Journal
c.schneider@embojournal.org

We realize that it is difficult to revise to a specific deadline. In the interest of protecting the conceptual advance provided by the work, we recommend a revision within 3 months (11th Aug 2025). Please discuss the revision progress ahead of this time with the editor if you require more time to complete the revisions. Use the link below to submit your revision:

Referee #1:

The bacterial AAA+ protease ClpC/ClpP is a key factor in bacterial stress responses and virulence. In this study Timo et al. explore the allosteric regulation, specifically, the role of the N-terminal domain (NTD) in modulating the ClpC activity. They demonstrate that the NTD is a negative regulator of ClpC, as its deletion stimulated the ATPase activity. Using a series of biophysical and biochemical tools, they show that the NTD stabilizes ClpC's resting inactive state. They further show that phosphoarginine (pArg) binding enhances ClpC's activity and mutating the pArg-binding sites destabilizes the resting state.

The authors also found by cryo-EM that the NTD stabilizes three assemblies of the ClpC resting state, i.e., the 10-mer, 12-mer, and 14-mer. Mutations at key NTD interfaces with the AAA1 domain and the cis and trans middle domains revealed the importance of the NTD in stabilizing the resting state. And a constitutively active NTD deletion mutant is shown to be toxic in vivo. They also modeled the binding of a cyclic peptide to ClpC and proposed how the peptide may deregulate the ClpC1 activity in *Mycobacterium tuberculosis*. Overall, the conclusion is supported by a comprehensive set of structural and functional data. This study improves our understanding of the ClpC regulation. Several issues listed below should be addressed before publication.

Major concerns

1. The negative-stain EM shows that in the absence of the dN domain, ClpC still interacts with ClpP and mediates the MD-MD interactions. However, in the $\Delta N/F436A$ double mutant, ClpC interacts with ClpP but fails to mediate the MD-MD interactions. It would be helpful to examine the impact of the F436A single mutation. Wild-type ClpC should be included as a control to show the expected decamer in the resting state under these conditions.
2. In Supplementary Figure 2A, the hexamer bands for ΔN and $\Delta N-F436A$ migrate more slower than the larger WT, WT+MecA, and F436A. Please explain this discrepancy. Please clarify if cross-linking was used in the DLS measurements and negative staining EM.
3. On page 6, the authors state, "ClpC-WT formed high molecular weight crosslink products, which hardly migrated into the SDS-gel, indicating resting state formation." Because the SDS-gel does not include a molecular weight marker above 180 kDa, it is unclear how the authors determined these high molecular weight species are hexameric and represent the resting state.
4. The authors should include the SEC profile in Fig. 2B to show the size distribution of the protein complexes. A standard calibration curve should be included to estimate the molecular weight of the eluted species. This would help validate the oligomeric states.
5. On page 10, the authors state that "pArg binding to such sites abrogates these interactions causing destabilization of the resting state." This is an important assertion. The authors should include EM data of the pArg-bound ClpC-WT to visualize the disruption and solidify the claim.
6. Fig. 4C shows two ClpC ring dimers, but it is unclear how this assembly structurally differs from the known resting ClpC decamer. A side-by-side comparison or a brief explanation is needed.
7. The authors describe three assemblies of the ClpC-WT resting state -10-mer, 12-mer, and 14-mer. Please discuss if they believe all are physiologically relevant or some might be in vitro artifact.
8. The refinement statistics for the 10-mer and 12-mer ClpC structures are missing in Table S1 and should be included.
9. It is unclear if the NTD residues E32 and E106 mediate any interactions with AAA1, MD, or MD*, and if E32 and E106 are around the cyclic peptide binding pocket.

Minor concerns

1. Consider including a sketch (perhaps as Fig. 1) to summarize existing and relevant knowledge on ClpC, such as (1) the known inactive, decameric resting state, (2) MecA binding to the NTD and MD, (3) the two pArg binding sites in the NTD, (4) the pArg-substrate-induced tetrahedral assembly, and (5) the supramolecular ClpC complex induced by CymA. Label the NTD and MD clearly in all relevant figures.
2. Page 7: "The formation of Δ N-ClpC ring dimers is reminiscent of ClpL ring dimers, which form by head-to-head MD-MD interactions." Please include a supplementary figure to compare this and highlight the similarity in the two MD-MD interactions.
3. Because F436 plays a critical role in mediating the MD-MD interactions, the F436A mutant should be included in Fig. 6B to compare with the NTD-AAA1, NTD-MD*, and NTD-MD mutants, in addition to the presence or absence of MecA.
4. In Table S1 please specify the unmasked reconstruction resolution of 7.7 Å corresponds to the FSC threshold of 0.5.

Referee #2:

In this manuscript, Jenne et al. investigate the mechanisms that underlie the allosteric control of the *S. aureus* ClpC/ClpP protease. Using a multifaceted approach of various biochemical, biophysical, and structural techniques, combined with specific deletions and point mutations, the authors elucidate the role of ClpC's N-terminal domains (NTD's) in stabilizing the resting state with spiral arrangements of 10, 12, or 14 subunits through inter-NTD, M-domain, and ATPase-domain interactions. These studies also convincingly reveal how the binding of adaptor proteins like MecA or of substrates containing phosphorylated Arg and hydrophobic regions disrupt those interactions and facilitate the transition to the active hexameric state of ClpC for ClpP binding and substrate degradation. With additional studies on *M. tuberculosis* ClpC1/ClpP, the authors confirm the conservation of those NTD-mediated mechanisms for ClpC activation and uncover the details of how cyclic peptides like CymA deregulate protease activity.

The presented structural and functional data are of good quality and nicely support the presented conclusions. This manuscript therefore provides an important advance to our understanding of bacterial ClpC/ClpP regulation and is well suited for publication in EMBO, after the rather minor points listed below have been addressed.

1) In addition to canonical single- or double-capped protease particles, the authors also observed larger complexes for Δ N-ClpC-F436A/ClpP and propose that those assemblies possibly originate from the recognition of other protease complexes as substrates. Although not ruled out, this model seems a bit surprising, given that proteases like ClpC/ClpP usually lack extended flexible and/or exposed hydrophobic regions in order to prevent autodegradation. Is it conceivable that the NTD deletion left behind an engageable tail on the N-terminus of ClpC?

The authors observe that addition of FITC-casein prevents this formation of larger complexes. However, considering the unfolded character and hydrophobicity of FITC-casein, the authors may want to rule out that this effect is primarily caused by non-specific interactions or changes in buffer conditions, similar to adding BSA. It may therefore be worth testing whether the same effect is observed when adding a more specific substrate like GFP-ssrA, which is robustly recognized and degraded by Δ N-ClpC-F436A/ClpP and should therefore also prevent the formation of larger assemblies.

2) When characterizing the role of the 2 phospho-arginine binding pockets for ClpC activation, the authors observed that the presence of pArg allows wild-type ClpC/ClpP to degrade FITC-casein at a similar rate as MecA-activated ClpC/ClpP. Given that GFP-ssrA is a more demanding substrate whose access to the central pore is blocked by NTDs in the absence of MecA, even when M-domain interactions are disrupted through the F436A mutation, it would be informative to assess whether pArg binding to NTDs is sufficient for GFP-ssrA degradation. This would directly address whether pArg binding opens access to the central pore and whether a ssrA-tagged substrate that contains phosphorylated Arg, but lacks exposed hydrophobic regions, can get readily degraded even in the absence of MecA.

3) If possible, the authors should try to normalize the degradation of FITC-casein and GFP-ssrA based on the total signal and provide true degradation rates in min⁻¹, rather than a relative fluorescence gain in % min⁻¹. Also, it makes no sense to report negative degradation rates for GFP-ssrA in Fig. 1C, as the increase in GFP fluorescence is likely due to signal drift, noise, or non-specific binding, but certainly not due to GFP synthesis.

4) The authors observed a proteolytic activity for ClpC-D356A/ClpP that exceeded the activity of MecA-bound wild-type ClpC/ClpP, and, interestingly, the addition of MecA led to a partial inhibition. The authors may comment on whether MecA binding could partially interfere with FITC-casein binding for degradation?

Minor points:

- 1) The authors use the 3-letter and 1-letter codes for amino acid positions (e.g. on page 13), and should stick with just one nomenclature.
- 2) In the paragraph at the top of page 11, there is a duplication of the sentence "No other oligomeric intermediate was observed

in mass photometry or cryo-EM SPA, suggesting ...".

3) On page 14, the sentence "This tetrahedron state again explains the formation..." is a bit misleading, because the preceding sentence is not about a tetrahedron state and what it refers to is unclear (presumably the sentence before the previous).

4) 4th line on page 15, it should say ClpC1-D364A (not ClpC-D364A).

Response letter for EMBOJ-2025-120881. Comments of the reviewers are shown in italics.

Reviewer 1

Major concerns

1. The negative-stain EM shows that in the absence of the dN domain, ClpC still interacts with ClpP and mediates the MD-MD interactions. However, in the $\Delta N/F436A$ double mutant, ClpC interacts with ClpP but fails to mediate the MD-MD interactions. It would be helpful to examine the impact of the F436A single mutation. Wild-type ClpC should be included as a control to show the expected decamer in the resting state under these conditions.

We performed the EM analysis of ClpC-DWB and ClpC-F436A-DWB/ClpP complexes as requested. ClpC-DWB forms a resting state whereas ClpC-F436A/ClpP forms proteasome-like complexes. We added respective 2D class averages to Fig. 2C.

2. In Supplementary Figure 2A, the hexamer bands for ΔN and $\Delta N-F436A$ migrate more slower than the larger WT, WT+MecA, and F436A. Please explain this discrepancy. Please clarify if cross-linking was used in the DLS measurements and negative staining EM.

We consider the migrations of crosslinked hexamer bands as comparable. We provide another example for the reviewer below, documenting a similar running behaviour. We are aware that glutaraldehyde crosslinking is a rough, qualitative method, however, the obtained data are yet in full agreement with other methods that we used to determine the assembly states of ClpC variants, including EM, SEC and DLS. We did not use crosslinking reagents in DLS and negative staining EM experiments and added this clarification to the methods section.

Figure: Glutaraldehyde (GA) crosslinking of ClpC-WT and indicated mutants was performed in presence of ATP γ S. Crosslinking reactions were analyzed before (0 min) and after (2/10 min) GA addition by SDS-PAGE. Crosslink product identities are indicated. A protein standard (kDa) is provided.

3. On page 6, the authors state, "ClpC-WT formed high molecular weight crosslink products, which hardly migrated into the SDS-gel, indicating resting state formation." Because the SDS-gel does not include a molecular weight marker above 180 kDa, it is unclear how the authors determined these high molecular weight species are hexameric and represent the resting state.

We have used glutaraldehyde crosslinking before to validate hexamer formation of ClpB variants (Franke *et al*, 2017) as simple and fast readout and thereby can assess the position of a

crosslinked AAA+ hexamer. Indeed, we have shown before that crosslinked *S. aureus* ClpC and *M. tuberculosis* (Mtb) ClpC1 M-domain (MD) mutants, which are deficient in undergoing head-to-head MD-MD interactions and form hexamers, run at the same position as crosslinked ClpB hexamers serving as reference (see below) (Carroni *et al*, 2017; Taylor *et al*, 2022). The running behaviours of crosslinked ClpC WT or mutants are also in agreement with other assembly data acquired by a broad spectrum of methods as pointed out above.

Figure S1 from Taylor *et al*, JBC, 2022: Glutaraldehyde crosslinking of Mtb ClpC1-WT and ClpC1-F444S was performed in absence and presence of ATP γ S without and with CymA as indicated. Crosslinking reactions of *E. coli* ClpB and *S. aureus* (Sa) ClpC served as references for hexamer and resting state formation. Crosslink products were analyzed by SDS-PAGE.

4. The authors should include the SEC profile in Fig. 2B to show the size distribution of the protein complexes. A standard calibration curve should be included to estimate the molecular weight of the eluted species. This would help validate the oligomeric states.

The running buffer used in these SEC runs included 2 mM ATP. This high ATP concentration dominates the UV signal and the elution profiles of protein complexes can thus not be traced. We therefore provide the elution profiles as SDS-PAGE. The structures of the ClpC resting state and the protease complexes (e.g. ClpC/MecA/ClpP, ClpC-F436A/ClpP) are extended but not globular and their molecular weights thus cannot be correctly determined by a calibration curves based on globular marker proteins. We would like to point out that we directly analyzed the structures of respective complexes by EM, providing direct visualization of their oligomeric states.

5. On page 10, the authors state that "pArg binding to such sites abrogates these interactions causing destabilization of the resting state." This is an important assertion. The authors should include EM data of the pArg-bound ClpC-WT to visualize the disruption and solidify the claim.

The addition of pArg is not sufficient to induce hexamer and tetrahedron (via MD-MD interactions) formation as shown in Fig. 4A. The additional presence of a substrate (e.g. FITC-casein) is required as additional trigger. A similar behavior was noticed for other ClpC NTD mutants (Fig. 4A, 6D) and we therefore conclude that NTD mutants or pArg binding destabilize the resting state without completely abrogating its formation.

We have performed the requested EM analysis, showing that pArg-bound ClpC still forms a resting state-like, non-hexameric structure, which, however, appears less homogenous and

regular as compared to the ClpC-WT resting state. A similar observation is made for e.g. ClpC-E32A/E106, suggesting that pArg binding or mutating the pArg1/2 sites impacts the organization of the ClpC resting state. This structural change likely destabilizes the resting state and allows for hexamerization in additional presence of substrate. We added respective EM data to Fig. 4C and Appendix Fig. S5B/C and discuss the implications of the noticed structural differences (page 10, lane 16-19).

6. Fig. 4C shows two ClpC ring dimers, but it is unclear how this assembly structurally differs from the known resting ClpC decamer. A side-by-side comparison or a brief explanation is needed.

We have performed a low-resolution 3D structure determination using the negative staining EM data of ClpC-WT+pArg+FITC-casein and of ClpC-WT alone. The 3D reconstructions show that the dimeric ClpC-WT+pArg+FITC-casein assembly is composed of planar rings, while the ClpC-WT resting state represents a pitched assembly (Appendix Fig. S5E). These findings confirm the different structural organizations of ClpC ring dimers found in presence of pArg and substrate FITC-casein and the ClpC resting state.

We have now quantified the frequency of dimer formation (approx. 25% ClpC dimers compared to 75% ClpC tetrahedrons) and added respective numbers (page 10, lane 21).

7. The authors describe three assemblies of the ClpC-WT resting state -10-mer, 12-mer, and 14-mer. Please discuss if they believe all are physiologically relevant or some might be in vitro artifact.

Mass photometry and cryoEM analysis reveal that ClpC forms the indicated diverse resting states. We believe that the diverse states are physiologically relevant and represent states of association and dissociation of ClpC dimers. We consider ClpC dimers as unstable and they will dissociate into monomers as also observed in mass photometry (Fig. 5A). Monomerization will make all allosteric NTD and MD sites available for substrate or adaptor binding, ultimately inducing ClpC activation. We discuss the relevance of the diverse resting states in the revised manuscript (page 18, lane 8-13). A validation of ClpC resting state heterogeneity *in vivo* demands cryo electron tomography analysis. While we consider such analysis highly relevant, we believe that it is technically very challenging and far beyond the scope of this study.

8. The refinement statistics for the 10-mer and 12-mer ClpC structures are missing in Table S1 and should be included.

We added the refinement statistics to Table S1.

9. It is unclear if the NTD residues E32 and E106 mediate any interactions with AAA1, MD, or MD, and if E32 and E106 are around the cyclic peptide binding pocket.*

The pArg1/2 sites are not engaged in binding of cyclic peptides, which bind to the hydrophobic groove of the NTD. The residues E32 and E106 are not in direct contact with AAA1, MD or MD*, yet binding of pArg to these sites clashes with NTD-MD and NTD-MD* interactions that stabilize the resting state (Fig. EV5A). The impact of E32A and E106A mutants on resting state formation is therefore indirect, likely stemming from local structural alterations of the pArg1/2 sites. We made a respective comment in the text (page 17, lane 15-18). We have additionally analyzed the role of T7, which is part of the pArg1 site that we identified as most relevant. T7 contacts the MD* residue E426. Assembly and activity profiles of the ClpC-T7D

mutant strongly resemble those of the pArg1 mutant E106A and other activating NTD mutants: (i) increased ATPase activity, (ii) autonomous proteolytic activity towards FITC-casein and (iii) formation of tetrahedrons in presence of substrate FITC-casein. We added an additional cartoon illustrating the T7-E426 contact as new Appendix Fig. S9A and the respective biochemical data to Appendix Fig. S9B-D. Notably, a partial activation of a ClpC-E426A mutant was noticed before (Carroni *et al.*, 2017), validating that the interactions between the pArg1 site and MD* are crucial for resting state stability.

Minor concerns

1. Consider including a sketch (perhaps as Fig. 1) to summarize existing and relevant knowledge on ClpC, such as (1) the known inactive, decameric resting state, (2) MecA binding to the NTD) and MD, (3) the two pArg binding sites in the NTD, (4) the pArg-substrate-induced tetrahedral assembly, and (5) the supramolecular ClpC complex induced by CymA. Label the NTD and MD clearly in all relevant figures.

We followed the suggestion of the reviewer and show published key ClpC assembly states with labeled NTDs and MDs in Fig. 1A. Additionally, we changed the color of the column bars representing the NTD-deletion mutant (Fig. 1B/D) accordingly to match the blue color of the NTD introduced in the new Figure 1A.

2. Page 7: "The formation of ΔN -ClpC ring dimers is reminiscent of ClpL ring dimers, which form by head-to-head MD-MD interactions." Please include a supplementary figure to compare this and highlight the similarity in the two MD-MD interactions.

We did not determine a high resolution structure of ΔN -ClpC dimers, thus a direct comparison to the cryoEM structure of a ClpL dimer (Kim *et al.*, 2020) is not possible. We provide the ClpL ring dimer structure as Appendix Fig. S3B, highlighting the residue F350, which is crucial for dimer formation via MD-MD interactions. The homologous residue F436 in case of ClpC is also essential for dimer formation of ΔN -ClpC rings (new sequence alignment in Appendix Fig. S3C). We added this information to the revised manuscript (page 7, lane 32).

3. Because F436 plays a critical role in mediating the MD-MD interactions, the F436A mutant should be included in Fig. 6B to compare with the NTD-AAA1, NTD-MD*, and NTD-MD mutants, in addition to the presence or absence of MecA.

We added ClpC-F436A data to Fig. 6B. These data were only acquired in absence of MecA since the mutation abolishes MecA interaction (Wang *et al.*, 2011).

4. In Table S1 please specify the unmasked reconstruction resolution of 7.7 Å corresponds to the FSC threshold of 0.5.

We revised Table S1 accordingly.

Reviewer 2

Minor points

1) In addition to canonical single- or double-capped protease particles, the authors also observed larger complexes for ΔN -ClpC-F436A/ClpP and propose that those assemblies possibly originate from the recognition of other protease complexes as substrates. Although not ruled out, this model seems a bit surprising, given that proteases like ClpC/ClpP usually

lack extended flexible and/or exposed hydrophobic regions in order to prevent autodegradation. Is it conceivable that the NTD deletion left behind an engageable tail on the N-terminus of ClpC?

The authors observe that addition of FITC-casein prevents this formation of larger complexes. However, considering the unfolded character and hydrophobicity of FITC-casein, the authors may want to rule out that this effect is primarily caused by non-specific interactions or changes in buffer conditions, similar to adding BSA. It may therefore be worth testing whether the same effect is observed when adding a more specific substrate like GFP-ssrA, which is robustly recognized and degraded by deltaN-ClpC-F436A/ClpP and should therefore also prevent the formation of larger assemblies.

Our Δ N-ClpC construct retains a flexible, thirteen residue long linker sequence (147MSNKNAQASKSNN159) that might be recognized as substrate-like tail. We followed the suggestion of the reviewer and performed SEC runs with either BSA or GFP-SsrA (see figure below). GFP-SsrA or BSA addition hardly reduced the formation of large Δ N-ClpC-F436A/ClpP complexes, indicating that the effect noticed before is specific for FITC-casein. One would also have expected an effect of GFP-SsrA, the differences to FITC-casein may be due to differences in affinities or binding modes. We added the information on the N-terminal tail to the manuscript and partially modified the statement on self-recognition (page 8, lane 6-9).

Figure: Superose6 SEC analysis of Δ N-ClpC-F436A-DWB/ClpP complexes in absence (ctrl) or presence of GFP-SsrA or BSA. Elution fractions were analyzed by SDS-PAGE followed by silver staining.

2) When characterizing the role of the 2 phospho-arginine binding pockets for ClpC activation, the authors observed that the presence of pArg allows wild-type ClpC/ClpP to degrade FITC-casein at a similar rate as MecaA-activated ClpC/ClpP. Given that GFP-ssrA is a more demanding substrate whose access to the central pore is blocked by NTDs in the absence of MecaA, even when M-domain interactions are disrupted through the F436A mutation, it would be informative to assess whether pArg binding to NTDs is sufficient for GFP-ssrA degradation. This would directly address whether pArg binding opens access to the central pore and whether a ssrA-tagged substrate that contains phosphorylated Arg, but lacks exposed hydrophobic regions, can get readily degraded even in the absence of MecaA.

We probed for degradation of GFP-SsrA by either pArg-activated ClpC/ClpP or by ClpC-F436A in presence of pArg (Appendix Fig. S4C). For both conditions we did not observe GFP-SsrA proteolysis. This finding implies that occupation of the pArg1/2 binding pockets is not sufficient to overcome the repressing function of the NTD in case of ClpC-F436A. Instead, binding of a hydrophobic substrate stretch to the hydrophobic groove of the NTD seems necessary. The same reason might apply for ClpC-WT. We conclude, that ClpC activation upon occupation of pArg1/2 sites requires the additional binding of a hydrophobic segment to the NTD groove. We made a respective comment in the text of the revised manuscript (page 9, lane 6-11).

3) If possible, the authors should try to normalize the degradation of FITC-casein and GFP-ssrA based on the total signal and provide true degradation rates in min⁻¹, rather than a relative fluorescence gain in % min⁻¹.

Also, it makes no sense to report negative degradation rates for GFP-ssrA in Fig. 1C, as the increase in GFP fluorescence is likely due to signal drift, noise, or non-specific binding, but certainly not due to GFP synthesis.

Fig. 1C: We agree with the reviewer and revised the figure accordingly. We now subtract the signal change in control reactions monitoring GFP-SsrA fluorescence in presence of ClpP only.

The readout % fluorescence decrease/min directly reports on GFP-SsrA degradation. In case of FITC-casein we agree with the reviewer that the % fluorescence increase/min value does not represent a true degradation rate. We, however, prefer keeping the fluorescence-based value as it directly correlates with proteolytic activities. We also consider its quantification more accurate as compared to an SDS-gel based assay. We made a comment in the revised text stating that the shown quantifications of FITC-casein fluorescence do not represent total degradation rates (page 5, lane 23-25).

4) The authors observed a proteolytic activity for ClpC-D356A/ClpP that exceeded the activity of MecA-bound wild-type ClpC/ClpP, and, interestingly, the addition of MecA led to a partial inhibition. The authors may comment on whether MecA binding could partially interfere with FITC-casein binding for degradation?

MecA is functioning as an adaptor but also as substrate of ClpC and thus targets itself for ClpC/ClpP-mediated degradation. This dual role of MecA has two consequences: First, it limits ClpC activation to short time periods and second, MecA competes with substrates for degradation. This competition can lower substrate degradation rates as observed here for ClpC-D356A, which still binds MecA. We clarify this point in the revised manuscript (page 14, lane 10-12).

Minor points

1) The authors use the 3-letter and 1-letter codes for amino acid positions (e.g. on page 13), and should stick with just one nomenclature.

We now only use the 1-letter code for amino acids.

2) In the paragraph at the top of page 11, there is a duplication of the sentence "No other oligomeric intermediate was observed in mass photometry or cryo-EM SPA, suggesting ...".

We corrected the error.

3) On page 14, the sentence "This tetrahedron state again explains the formation..." is a bit misleading, because the preceding sentence is not about a tetrahedron state and what it refers to is unclear (presumably the sentence before the previous).

We changed the text to: "EM analysis documents that ClpC-D356A forms a resting state-like structure in absence of substrate (Fig. 6E, Appendix Fig. S9G). Addition of FITC-casein triggers formation of hexamer-based ClpC-D356A assemblies, largely representing tetramers of ClpC hexamers, but also assemblies consisting of at least five ClpC hexamers (Fig. 6E, Appendix Fig. S9G). These assembly states again explain the formation of very large, heterogenous assemblies of FITC-casein bound ClpC-D356A in additional presence of ClpP (Fig. 6D)."

4) 4th line on page 15, it should say *ClpC1-D364A* (not *ClpC-D364A*).

We corrected the mistake.

References

- Carroni M, Franke KB, Maurer M, Jager J, Hantke I, Gloge F, Linder D, Gremer S, Turgay K, Bukau B *et al* (2017) Regulatory coiled-coil domains promote head-to-head assemblies of AAA+ chaperones essential for tunable activity control. *eLife* 6
- Franke KB, Bukau B, Mogk A (2017) Mutant Analysis Reveals Allosteric Regulation of ClpB Disaggregase. *Front Mol Biosci* 4: 6
- Kim G, Lee SG, Han S, Jung J, Jeong HS, Hyun JK, Rhee DK, Kim HM, Lee S (2020) ClpL is a functionally active tetradecameric AAA+ chaperone, distinct from hexameric/dodecameric ones. *FASEB J* 34: 14353-14370
- Taylor G, Frommherz Y, Katikaridis P, Layer D, Sinning I, Carroni M, Weber-Ban E, Mogk A (2022) Antibacterial peptide CyclomarinA creates toxicity by deregulating the Mycobacterium tuberculosis ClpC1-ClpP1P2 protease. *J Biol Chem* 298: 102202
- Wang F, Mei Z, Qi Y, Yan C, Hu Q, Wang J, Shi Y (2011) Structure and mechanism of the hexameric MecA-ClpC molecular machine. *Nature* 471: 331-335

Dear Dr. Mogk,

Thank you for submitting a revised version of your manuscript. Your study has now been seen by all original referees, who find that their previous concerns have been addressed and now recommend publication of the manuscript. There remain only a few mainly editorial points that have to be addressed before I can extend formal acceptance of the manuscript:

- Please rename the Conflict of Interest section into "Disclosure and Competing Interests Statement", in accordance with our updated Guide to Authors (<https://www.embopress.org/competing-interests>)
- As we are switching from a free-text author contribution statement towards a more formal statement based on Contributor Role Taxonomy (CRediT) terms, please remove the present Author Contribution section and instead specify each author's contribution(s) directly in the Author Information page of our submission system during upload of the final manuscript. See <https://casrai.org/credit/> for more information.
- Please rename the callouts for Table S1-S2 to Appendix Table S1-S2
- Please complete the general info table in the author Checklist
- Please renumber the EV figures to Figure EV1-EV2 with the corresponding callouts and correct labels in figure legends (Figure EV1-EV2 instead of Expanded View Figure 4-5)
- The title page of the "APPENDIX 1 FILE WITH Table of Content" should contain "Appendix for + ms title"; Appendix figure legends should be below the corresponding figures, and above the Appendix tables, not listed together; change the Appendix reference list from numbered to ordered-by-author according to our guidelines with 10 authors + et al.
- Please provide suggestions for a short 'blurb' text prefacing and summing up the conceptual aspect of the study in two sentences (max. 250 characters), followed by 3-5 one-sentence 'bullet points' with brief factual statements of key results of the paper; they will form the basis of an editor-written 'Synopsis' accompanying the online version of the article. Please also provide an altered synopsis image, making sure that the aspect ratio conforms to our website's format - it should be exactly 550 pixels wide and between 300-600 pixels high.
- Please provide the specific URLs for EMD-53014, EMD-53324, EMD-53312, 9QCL, 9QRW and 9QQR datasets in the data availability statement.
- Figure Legends (main + EV):
 1. Please note that the exact p values are not provided in the legends of figures 2A, 4B
 2. Please note that the box plots need to be defined in terms of minima, maxima, centre, bounds of box and whiskers, and percentile in the legend of figure 6D
 3. Please note that information related to n is missing in the legend of figure 6D
 4. Please note that the measure of center for the error bars needs to be defined in the legends of figures 1B, C, D, E; 3A, B, D; 6B, C; 7B, C
- Please rename "Materials and methods" to "Methods"
- Please remove the legends of the 3 movie files from the manuscript and zip with each movie file.
- Please correct the sections names and order: Title page - Abstract - Keywords - Introduction - Results - Discussion - Methods - Data Availability - Acknowledgements - Disclosure and Competing Interests Statement - References - Figure Legends - Table(s) - Expanded View Figure Legends.

With best regards,

Cornelius Schneider

Cornelius Schneider, PhD
Editor | The EMBO Journal
c.schneider@embojournal.org

Please refer to our figure preparation guideline in order to ensure proper formatting and readability in print as well as on screen:

See also figure legend guidelines:

<https://www.embopress.org/page/journal/14602075/authorguide#figureformat>

Use the link below to submit your revision:

Referee #1:

The authors have done a great job fully addressing our concerns.

Referee #2:

The authors thoroughly considered the comments of both reviewers and appropriately addressed criticism with several additional experiments and revisions to the text. I am satisfied with the revised manuscript and I think it is now ready for publication in The EMBO Journal.

All editorial and formatting issues were resolved by the authors.

Dear Dr. Mogk,

I am pleased to inform you that your manuscript has been accepted for publication in the EMBO Journal.

Yours sincerely,

Cornelius Schneider, PhD
Editor
The EMBO Journal
c.schneider@embojournal.org
